# Positively charged specificity site in cyclin B1 is essential for mitotic fidelity

Christian Heinzle[1,2], Anna Höfler [3], Jun Yu[3], Peter Heid [1,2], Nora Kremer [4], Rebecca Schunk [1,2], Florian Stengel [1,2], Tanja Bange [4], Andreas Boland [3] ✉ & Thomas U. Mayer [1,2] ✉

Phosphorylation of substrates by cyclin-dependent kinases (CDKs) is the driving force of cell cycle progression. Several CDK-activating cyclins are involved, yet how they contribute to substrate specificity is still poorly understood. Here, we discover that a positively charged pocket in cyclin B1, which is exclusively conserved within B-type cyclins and binds phosphorylated serine- or threonine-residues, is essential for correct execution of mitosis. HeLa cells expressing pocket mutant cyclin B1 are strongly delayed in anaphase onset due to multiple defects in mitotic spindle function and timely activation of the E3 ligase APC/C. Pocket integrity is essential for APC/C phosphorylation particularly at non-consensus CDK1 sites and full in vitro ubiquitylation activity. Our results support a model in which cyclin B1's pocket facilitates sequential substrate phosphorylations involving initial priming events that assist subsequent pocket-dependent phosphorylations even at non-consensus CDK1 motifs.

For the survival of every organism, it is essential that the different cell cycle events are executed not only with the highest possible fidelity, but also in the correct order. Key pacemakers of cell cycle progression are cyclin-dependent kinases (CDKs), which in complex with activating cyclin subunits control the timing of cell cycle events by phosphorylating distinct substrates in the right temporal order[1–3]. The spectrum of substrates involves hundreds of proteins, whose localizations, functions, or stabilities are altered upon phosphorylation by CDK/cyclin[4–6]. Deregulated cell cycle control is involved in the pathogenesis of cancers demonstrating the importance of tightly regulated CDK/cyclin activities[7]. Despite their central role in accurate cell cycle progression, we still have only an incomplete understanding of how CDK/cyclin complexes recognize and phosphorylate their substrates at the right time during the cell cycle.

At the biochemical level, substrate phosphorylation by CDK/cyclin is driven by three main interactions. First, the CDK active site has a preference for proline at position +1 and a basic residue at position +3 giving rise to a (S/T)Px(K/R) consensus motif (x = any amino acid)[8,9]. Second, cyclins bind short linear motifs (SLiMs) in substrates via a hydrophobic patch region within the evolutionarily conserved cyclin box[10–14]. In yeast, known SLiMs are the LP motif (G1 cyclins), RxL motif (S and G2 cyclins), and LxF motif (M cyclins) and a comparable differential utilization of docking motifs has been anticipated for mammalian cyclins[15–21]. Third, CKS1 or CKS2 (CDK subunit) proteins associate with CDK/cyclin complexes and bind phosphorylated threonine residues in primed substrates[22,23]. By acting as phosphate-adaptor proteins, CKS1/CKS2 confer substrate specificity and facilitate processive multisite phosphorylations even at non-proline-directed CDK sites[24]. In addition to its kinase function, CDK1/cyclin B acts as a stochiometric inhibitor of separase, a cysteine endopeptidase that cleaves the cohesin ring encircling sister chromatids[25]. Specifically, phosphorylation of human separase serine-1126 (pS1126) by CDK1 promotes complex formation between CDK1/cyclin B and separase resulting in mutual inhibition of both enzymes[26]. At anaphase onset, separase activated by the destruction of its inhibitors securin[27] and cyclin B[28] cleaves the cohesin protein Rad21 (also known as Scc1), resulting in the liberation of sister chromatids and their segregation into nascent daughter cells.

[1]Department of Biology, University of Konstanz, Konstanz, Germany. [2]Konstanz Research School Chemical Biology, University of Konstanz, Konstanz, Germany. [3]Department of Molecular and Cellular Biology, University of Geneva, Geneva, Switzerland. [4]Institute of Medical Psychology and Biomedical Center (BMC), Faculty of Medicine, LMU, Munich, Germany. ✉e-mail: Andreas.Boland@unige.ch; Thomas.U.Mayer@uni-konstanz.de

Recently, the structure of human CDK1/CKS1/cyclin B1 bound to separase was determined by cryogenic electron microscopy (cryoEM)[29]. This study identified a hitherto unknown phosphate-binding pocket (PBP) in cyclin B1 with arginine-307 (R307), histidine-320 (H320), and lysine-324 (K324) forming a hydrogen-bonding network with the phosphate group attached to S1126 of separase (Fig. 1A). Notably, critical residues of the positively charged PBP are conserved within B-type, but not A-type cyclins (Fig. 1B). The absence of this pocket in A-type cyclins explains why cyclin B1 and B2, but not cyclin

A2, bind to and inhibit separase[30,31]. Consistent with the essential function of pS1126 for cyclin B1 binding, separase with a non-phosphorylatable S1126A mutation is resistant to cyclin B1-mediated inhibition[28]. For budding yeast, it has been shown that the PBP of the cyclin B homolog Clb2 contributes together with CKS1 to multisite phosphorylation of the transcription activator Ndd1 resulting in its degradation during prolonged mitotic arrest[32]. Thus, Clb2 uses its PBP to mediate interactions with pre-phosphorylated substrates facilitating sequential phosphorylation events, a function previously

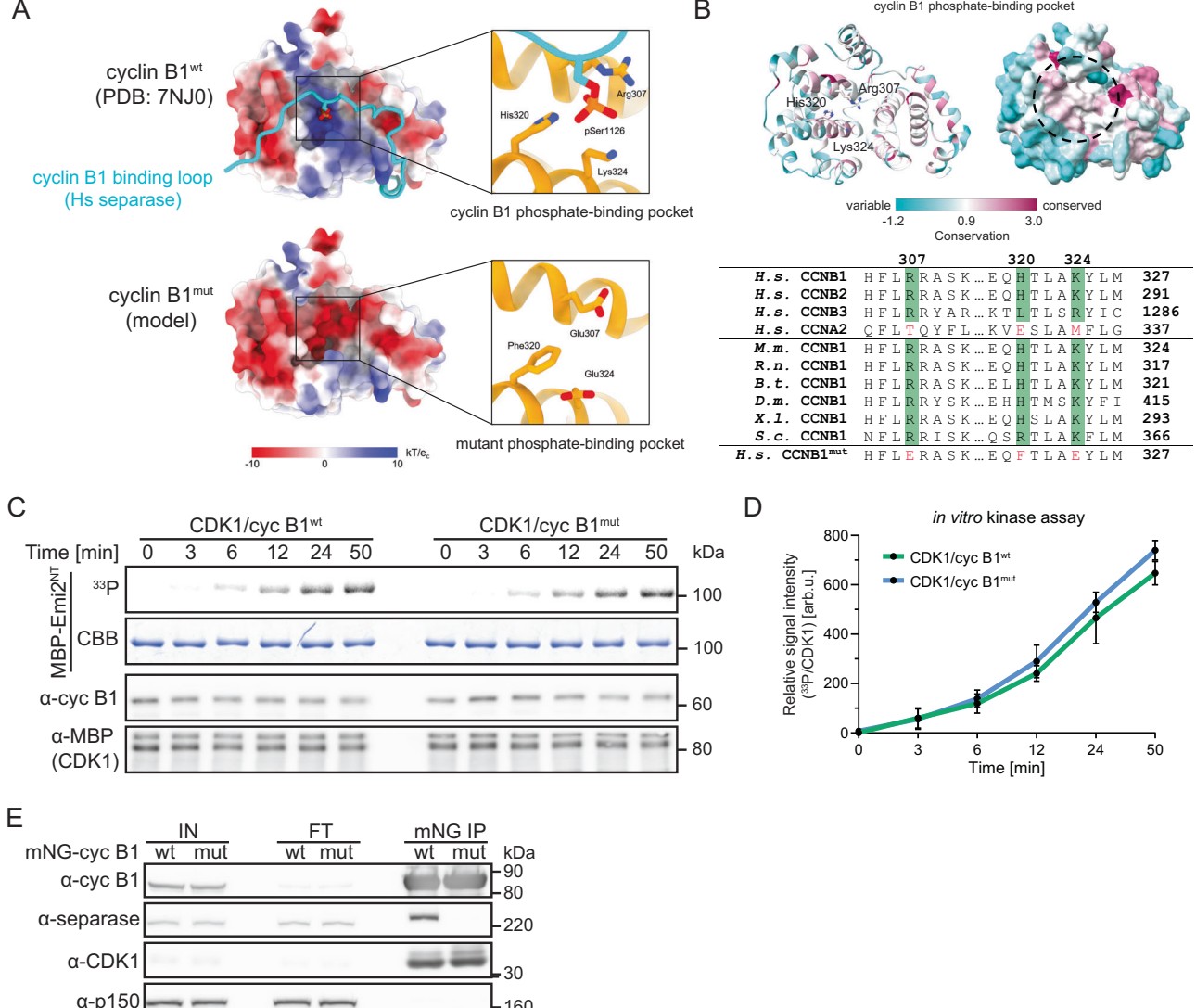

**Fig. 1 | Cyclin B1^mut is capable of activating CDK1, but deficient in separase binding. A** Electrostatic surface potentials of human cyclin B1^wt (top, PDB: 7NJ0) and cyclin B1^mut (bottom, modeled) are illustrated. Close-up views of the cyclin B1 phosphate-binding pocket in cyclin B1^wt and in cyclin B1^mut bound to phosphoserine 1126 of human separase. Mutations of three critical residues (R307E, H320F, K324E) in the binding pocket lead to a complete charge reversal. Electrostatic potentials are contoured from -10 (red) to +10 kTe^-1 (blue). **B** Sequence conservation of cyclin B1 mapped onto the human cyclin B1 structure (PDB: 7NJ0)[29]. The cyclin B1 structure is shown as ribbon (left) or surface representation (right). Residues that form the phosphate-binding pocket are displayed as sticks in the ribbon representation, and the phosphate-binding pocket is indicated with a dashed circle on the surface representation. The AL2CO program[89] has been used to map the conservation indices from a multiple sequence alignment of 2000 different cyclin B homologs (including yeast and human) onto the spatial structure of human cyclin B1. Variable regions are shown in cyan, conserved regions in maroon. Bottom, sequence alignment of the pocket region for different cyclins. Sequence

comparison of human B-type cyclins with cyclin A2 (top four rows). Sequence conservation of the binding pocket in B1-cyclins across different species (middle six rows). Last row shows the sequence of the designed cyclin B1^mut pocket. Residue numbers for each species are labeled on the right. *H.s.* (*Homo sapiens*), *M.m.* (*Mus musculus*), *R.n.* (*Rattus norvegicus*), *B.t.* (*Bos taurus*), *D.m.* (*Drosophila melanogaster*), *X.l.* (*Xenopus laevis*), *S.c.* (*Saccharomyces cerevisiae*). **C** Radioactive in vitro kinase assay using MBP-CDK1/cyclin B1^wt/mut-His and as substrate MBP-Emi2^NT (aa 1-350) containing a single CDK1 consensus site. At indicated timepoints after reaction start, samples were immunoblotted for MBP and cyclin B1 (α-cyc B1) and analyzed by autoradiography (^33P). CBB (Coomassie brilliant blue). **D** Quantification of n = 3 independent in vitro kinase assays. Shown are mean values with standard deviations. **E** Immunoblot analyses of anti-mNG immunoprecipitation (IP) samples of HeLa cells expressing mNG-cyclin B1^wt/mut. Doxycycline treated cells were enriched in mitosis by treatment with 333 nM nocodazole for 20 h. Shown are representative input (IN), flow through (FT) and IP samples. p150 served as IP control.

exclusively attributed to CKS proteins[22]. For Xenopus cyclin B2, it has been shown that the RRASK motif, which is part of the PBP, is important for binding to Cdc25C, a key CDK1 regulator[33].

Based on these data, we speculated that the pocket by binding to pre-phosphorylated substrates could serve as a specificity site that facilitates subsequent phosphorylation events in a strictly B-type cyclin specific manner. Such a pocket-dependent substrate anchoring mechanism could enhance phosphorylation of low-affinity sites leading to high-occupancy substrate phosphorylation and contribute to ultrasensitive switches in protein function. As shown previously[34], switch-like responses in protein function are critical for a timely and ordered execution of cell cycle processes.

## Results

### Cyclin B1^mut is deficient in separase binding, but not in CDK1 activation

To test our hypothesis that B-type cyclins employ their PBP to bind pre-phosphorylated substrates, we first created a PBP mutant version of cyclin B1 (cyclin B1^mut). To this end, we replaced the residues that form a critical hydrogen-bonding network with separase pS1126 with oppositely charged glutamates (R307E, K324E) and a phenylalanine (H320F) to maintain the aromatic character but eliminate the positive charge (Fig. 1A, B). Next, we performed in vitro kinase assays to confirm that the introduced mutations did not unspecifically affect CDK1/cyclin B1 activity by disturbing the fold of cyclin B1. For this assay, we purified a maltose-binding protein (MBP)-tagged fragment of Emi2 (MBP-Emi2^NT) containing a single CDK1 phosphorylation site and MBP-tagged CDK1 co-expressed with His-tagged wildtype (wt) or mutant (mut) cyclin B1. Indeed, CDK1/cyclin B1^mut was as efficient as CDK1/cyclin B1^wt in phosphorylating the Emi2 fragment (Fig. 1C, D). We further confirmed by size exclusion chromatography (SEC), circular dichroism (CD) spectroscopy, thermal shift assays, and in vitro pulldown assays that cyclin B1^mut behaved like cyclin B1^wt and was competent to form a complex with CDK1/CKS1 (Fig. S1A–D).

After having validated cyclin B1^mut, we created stable HeLa cell lines that upon doxycycline induction (+ dox) expressed mNeonGreen (mNG)-tagged wt or mutant cyclin B1 (HeLa mNG-cyclin B1^wt/mut). Doxycycline-treated cells were enriched in mitosis by nocodazole treatment, harvested by shake-off, and α-mNG immunoprecipitation (IP) was performed from cell lysates. As expected, Western blotting (WB) revealed that endogenous CDK1 co-precipitated equally well with wt and mutant cyclin B1. In contrast, separase only efficiently co-precipitated with cyclin B1^wt but not cyclin B1^mut (Fig. 1E). In vitro pulldown assays using recombinant proteins confirmed the inability of cyclin B1^mut to bind separase (Fig. S1E). Notably, separase with a threonine mutation at position 1126 (separase^S1126T), when in vitro phosphorylated, and even with a phosphate-mimicking mutation (separase^S1126E) bound to purified CDK1/CKS1/cyclin B1 (CCC), indicated by a shift towards higher molecular weight in SEC compared to separase or the CCC complex alone (Fig. S1F). From these data, we concluded that the pocket accommodates phosphorylated serine and threonine, as well as negatively charged residues. In sum, these data confirmed that the introduced mutations abolish cyclin B1's ability to bind phosphorylated separase, without affecting its overall fold and function as activating subunit for CDK1.

### Cells expressing cyclin B1^mut display mitotic defects

Next, we investigated if the integrity of cyclin B1's PBP is critical for mitotic fidelity. Synchronized HeLa mNG-cyclin B1^wt/mut cells were depleted of endogenous cyclin B1 and treated with doxycycline to express ectopic, siRNA-resistant cyclin B1 (Fig. 2A). Of note, cyclin B2 was co-depleted (siB1&B2) to prevent that it compensates for loss of cyclin B1[35–37]. WB analyses after thymidine release confirmed efficient depletion of both cyclins in siB1&B2 cells compared to control siRNA-treated cells and expression of cyclin B1^wt/mut at endogenous levels

(Figs. 2B and S2A). Constitutive co-expression of mCherry-tagged histone H2B allowed us to quantify mitotic timing by live-cell microscopy. Control-depleted mNG-cyclin B1^wt/mut cells not treated with doxycycline efficiently entered mitosis and proceeded from nuclear envelope breakdown (NEBD) to anaphase onset with similar kinetics (HeLa mNG-cyclin B1^wt: 40.5 ± 0.9 min; HeLa mNG-cyclin B1^mut: 38.0 ± 4.7 min) validating the generated cell lines (Figs. 2C and S2B). In contrast, the majority of double-depleted siB1&B2 cells not expressing ectopic cyclin B1 failed to enter mitosis (Fig. S2B), which is in line with some but not all previous studies on the requirement of cyclin B1 and B2 for mitotic entry[35–38]. The defect in mitotic entry was rescued by the expression of ectopic cyclin B1 with the wt protein being slightly more efficient than the mutant (Fig. S2B). Notably, anaphase onset was strongly delayed in double depleted cells expressing cyclin B1^mut compared to wt expressing cells (161.3 ± 27.4 vs. 81.6 ± 2.4 min, respectively, Fig. 2C). The same effect was observed when we expressed cyclin B1 fused at its C-terminus to enhanced green fluorescent protein (eGFP) in siB1&B2 cells (Fig. S2C). Thus, the observed phenotype was not due to the position of the tag interfering with cyclin B1 function.

To understand the cause for delayed anaphase onset, we first analyzed the localization of cyclin B1, which reportedly localizes to mitotic centrosomes, chromatin, and unattached kinetochores[39–42]. siB1&B2 mNG-cyclin B1^wt/mut cells were released from a double thymidine block, treated with doxycycline and the solvent control DMSO or the microtubule depolymerizer nocodazole and analyzed by immunofluorescence (IF) microscopy. As expected, in DMSO-treated cells, mNG-cyclin B1^wt localized to both spindle poles and kinetochores and nocodazole treatment caused its dissociation from spindle poles and enhanced kinetochore localization (Fig. 2E). In sharp contrast, mNG-cyclin B1^mut localized to spindle poles, but not kinetochores in DMSO-treated cells, and nocodazole treatment induced its loss from spindle poles but did not increase its kinetochore localization. The same localization pattern was observed for cyclin B1-eGFP (Fig. S2D). As shown previously[39,40,43], cyclin B1 is recruited to unattached outer kinetochores by the checkpoint protein Mad1, while cyclin B1 serves as the anchor for Mad1 at the kinetochore corona, a transient meshwork that assembles around the outer kinetochore and aids microtubule attachment and SAC signaling. To gain insights into the mechanism underlying the failure of cyclin B1^mut to localize to kinetochores, we first analyzed the interaction between cyclin B1 and Mad1 by co-IP experiments. WB analyses of α-mNG IP samples from siB1&B2 cells revealed reproducibly, but not significantly, reduced binding of endogenous Mad1 to mNG-cyclinB1^mut compared to cyclin B1^wt (Fig. S2E), which in part could explain the failure of cyclinB1^mut to localize to kinetochores. Next, we investigated the localization of Mad1 in siB1&B2 cells expressing mNG-cyclin B1^wt/mut by IF microscopy. Of note, cells were treated with nocodazole to ensure that kinetochores were unattached. As shown in Fig. 2E, under these conditions, wt but not mutant cyclin B1 localized to unattached kinetochores (Fig. S2F). Notably, quantitative analyses revealed a roughly 50% loss of the Mad1 signal at kinetochores in cyclin B1^mut cells compared to wt expressing cells (Fig. S2F). This observation is in line with the fact that cyclin B1 acts as the anchor for Mad1 at the kinetochore corona, but not at the outer kinetochore[40,44]. Since the outer kinetochore pool of Mad1 is reported to be sensitive to MPS1 activity[40,44], Mad1 remaining at kinetochores in cyclin B1^mut cells should be lost upon MPS1 inhibition. Indeed, treatment of siB1&B2 cells expressing cyclin B1^mut with the MPS1 inhibitor AZ3146 resulted in the almost complete loss of the Mad1 signal at kinetochores (Fig. S2F). The identical treatment of cells expressing cyclin B1^wt reduced the Mad1 signal to about half suggesting that under these conditions the MPS1-sensitive, outer kinetochore pool of Mad1, but not the one localizing to the corona, is lost. Thus, while further studies are required to understand the exact molecular mechanism of cyclin B1's kinetochore localization, these data

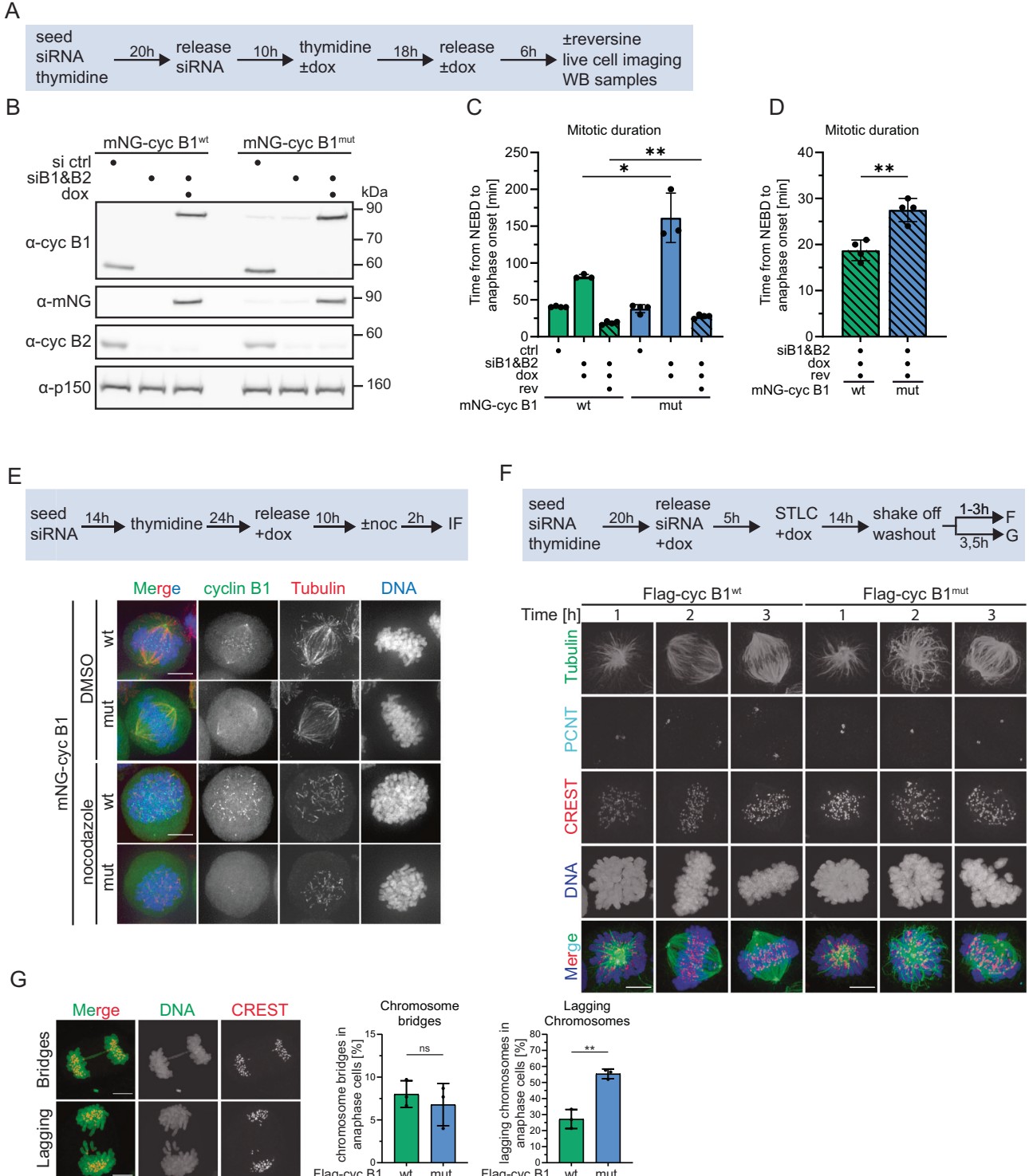

**Fig. 2 | Cyclin B1's phosphate-binding pocket is critical for mitotic fidelity.**
**A** Experimental outline to synchronize HeLa cells transfected with control (ctrl) or cyclin B1 and B2 siRNAs (siB1&B2) in S-phase via double thymidine block and to induce expression of mNG-cyclin B1$^{wt/mut}$ by doxycycline (dox) treatment.
**B** Immunoblot assessing depletion of endogenous cyclin B1 and B2 and expression of mNG-cyclin B1$^{wt/mut}$. p150 served as loading control. **C** Quantification of live-cell imaging. The mean of n = 3 or n = 4 independent experiments is shown with standard deviation. Unpaired t-test. **p = 0.002, *p = 0.0149. **D** Quantification shown in (**C**) with different y-axis scale for reversine conditions. Unpaired t-test**p = 0.002. **E** Representative pictures of mitotic mNG-cyclin B1$^{wt/mut}$ (green)

HeLa cells treated with either DMSO or nocodazole and stained for tubulin (red) and DNA (blue). Scale bar = 10 μm. **F** Timecourse to synchronize cells for STLC washout experiments. Images of representative cells processed at indicated timepoints, 1 to 3 h after STLC washout. Tubulin (green), pericentrin (PCNT) (cyan), CREST (red), and DNA (blue) are shown. Scale bar = 10 μm. **G** Representative images of siB1&B2 cells expressing mNG-cyclin B1$^{wt/mut}$ imaged 3.5 h post STLC release. Shown are examples of either lagging chromosomes or chromosome bridges with DNA and CREST in green and red, respectively. Right panel shows the quantification of both phenotypes. The mean with standard deviation of n = 3 independent experiments is shown. Unpaired t-test: **p = 0.0023.

suggested that the integrity of cyclin B1's phosphate-binding pocket is essential for cyclin B1's kinetochore localization and contributes to the correct localization of a distinct pool of Mad1.

Next, we investigated the process of bipolar spindle formation in stable cell lines that inducibly express (+ dox) Flag-tagged cyclin B1$^{wt/mut}$. For these studies, cells were analyzed that recovered from treatment with the small molecule STLC, which causes monopolar spindles by inhibiting Eg5[45]. After STLC washout, siB1&B2 Flag-cyclin B1$^{wt}$ cells efficiently established spindle bipolarity determined by pericentrin distance (Figs. 2F and S2H). Bipolar spindle formation in these cells was accompanied by loss of Mad1 from kinetochores indicating that these cells efficiently established proper microtubule attachments (Fig. S2G). In contrast, cells expressing Flag-cyclin B1$^{mut}$ were much slower in forming bipolar spindles that, in addition, often appeared misorganized (Figs. 2F and S2H). Consistently, in these cells, Mad1 dissociation was delayed compared to cyclin B1$^{wt}$ expressing cells indicative of malattached chromosomes (Fig. S2G).

Reportedly, defects in the proper attachment of chromosomes can result in lagging chromosomes upon anaphase onset[46]. Indeed, siB1&B2 cells expressing pocket mutant cyclin B1 displayed a higher frequency of lagging chromosomes, but not of chromosome bridges, upon anaphase onset following STLC washout (Fig. 2G). From these data we concluded that cells expressing cyclin B1$^{mut}$ have multiple defects in spindle formation and organization, which might be caused by mislocalized cyclin B1$^{mut}$ culminating in prolonged activation of the spindle assembly checkpoint (SAC). Ultimately, these defects result in chromosome segregation defects affecting mitotic fidelity. A corollary of this hypothesis is that SAC inactivation should abrogate the delay in anaphase onset observed in cyclin B1$^{mut}$ expressing cells. To test this, thymidine synchronized cells were released and treated with 1 µM reversine, an inhibitor of the SAC kinase MPS1[47]. Indeed, live-cell analyses revealed that reversine treatment greatly accelerated anaphase onset in siB1&B2-HeLa cyclin B1$^{mut}$ cells (Fig. 2C, D). However, timing of anaphase onset in these cells was still delayed compared to reversine-treated siB1&B2-HeLa cells expressing mNG-cyclin B1$^{wt}$ (Fig. 2D). Again, the same effect was observed for double depleted cells expressing cyclin B1-eGFP (Fig. S2C). Since reversine was added six hours after thymidine release, cells entered mitosis in the complete absence of a functional SAC. Thus, persistent anaphase delay of reversine treated cyclin B1$^{mut}$ cells suggested that these cells – irrespective of their spindle defects–most likely were unable to efficiently activate the APC/C. Indeed, WB analyses revealed that the degradation of the bona fide APC/C substrate securin was delayed in reversine-treated double depleted cells expressing cyclin B1$^{mut}$ cells compared to wt expressing cells (Fig. S3A). Thus, these data suggested that the integrity of cyclin B1's phosphate-binding pocket is critical for the correct function of the mitotic spindle affecting timely cell cycle progression.

## Cyclin B1's phosphate-binding pocket is important for APC/C activation

To understand the cause of the observed mitotic defects, we applied mass spectrometry (MS) to identify proteins that interact with cyclin B1 in a PBP-dependent manner. HeLa cells expressing Flag-cyclin B1$^{wt/mut}$ were treated with nocodazole to enrich for mitotic cells and harvested by shake-off. In three independent experiments, α-Flag IP was performed from uninduced HeLa Flag-cyclin B1$^{wt}$ cells (-dox), and cells expressing either Flag-cyclin B1$^{wt}$ or -cyclin B1$^{mut}$. MS analyses revealed that separase (ESPL1) was significantly enriched in the IP samples of cyclin B1$^{wt}$ expressing cells compared to cyclin B1$^{mut}$ and uninduced cells validating our MS-based proteomics approach (Fig. 3A). Other enriched proteins were inter alia the microtubule associated proteins pericentriolar material 1 (PCM1), G2 and S-phase expressed 1 (GTSE1), MgcRacGAP (RGAP1), the SAC proteins mitotic arrest deficient 2-like protein 1 (MAD2L1) and budding uninhibited by bezimidazoles (BUB1B). Notably, among the significantly enriched proteins were

several APC/C subunits. Since siB1&B2-HeLa mNG-cyclin B1$^{mut}$ cells were inefficient in APC/C activation under conditions when the SAC was inactive (Fig. 2D), we focused on the APC/C for further studies.

We first confirmed these results by WB analyses of Flag-cyclin B1 IP samples from mitotically arrested siB1&B2 cells. Consistent with being hyperphosphorylated in mitosis[48–50], in the input samples, APC3 was detectable as multiple smeared bands representing differentially phosphorylated APC3 forms (Fig. 3B). Notably, compared to cells expressing cyclin B1$^{wt}$, APC3 displayed faster mobility in the input samples of cells expressing cyclin B1$^{mut}$ (Fig. 3B). Thus, these data suggested that efficient phosphorylation of APC3 requires the integrity of cyclin B1's phosphate-binding pocket. APC3 co-precipitating with cyclin B1$^{mut}$ displayed faster SDS-PAGE mobility than the one associated with cyclin B1$^{wt}$. A potential cause could be that in the cyclin B1$^{mut}$ condition, phosphorylation sites are missing that mediate CDK1/cyclin B1 binding in a pocket-independent manner, e.g., via CKS1, or that the phosphorylation sites are present but cannot bind to pocket mutant cyclin B1. Lambda (λ) phosphatase treatment of the Flag-cyclin B1 IP bead samples resulted in the release of APC3 from either wt or mutant cyclin B1 confirming that the interaction is indeed phosphorylation-dependent (Fig. S3C).

To test if the phosphorylation of other APC/C subunits also depends on pocket integrity, we performed an anti-Flag IP from siB1&B2 HeLa cells stably expressing cyclin B1$^{wt/mut}$ and Flag-APC4. For these analyses, we used Phos-tag SDS-PAGE to enhance phosphorylation-dependent mobility changes. APC4 did not show any shift in its mobility irrespective of the Phos-tag conditions used. Notably, however, APC1 – like APC3 – showed enhanced mobility in the input and IP samples of cyclin B1$^{mut}$ cells compared to cyclin B1$^{wt}$ samples (Fig. S3B). In sum, from these data we concluded that cyclin B1's phosphate-binding pocket contributes to the association with the APC/C and efficient phosphorylation of APC3 as well as APC1 and potentially other APC/C subunits.

Next, we investigated how APC/C phosphorylation affects its ubiquitylation activity. Thymidine synchronized siB1&B2 mNG-cyclin B1$^{wt/mut}$ HeLa cells that constitutively expressed Flag-APC4 were released, harvested by mitotic shake-off, and treated for 30 min with reversine to exclude that differences in SAC activities between wt and mutant cyclin B1 cells affect APC/C activity (Fig. 3C). WB analyses of α-Flag-APC4 IP samples confirmed reversine-induced dissociation of Bub3 from the APC/C (Fig. S3D). Next, we performed in vitro ubiquitylation assays using α-Flag-APC4 IP samples and recombinant ubiquitin, UBA1 and E2 enzymes (UBE2C ± UBE2S) and as substrate a fluorescein-labeled N-terminal fragment of cyclin B1 (cyc B1$^{1–70}$). Fluorescent readout revealed that APC/C immunoprecipitated from mNG-cyclin B1$^{mut}$ expressing cells was indeed less efficient in cyc B1$^{1–70}$ ubiquitylation compared to cyclin B1$^{wt}$ expressing cells (Fig. 3D). The difference was most obvious for highly ubiquitylated species of cyc B1$^{1–70}$ and this effect was enhanced when both UBE2S and UBE2C were included. We concluded that cyclin B1's PBP is critical for efficient APC/C activation.

To confirm this finding, we performed in vitro ubiquitylation assays using APC/C immunopurified from Xenopus egg extract. Xenopus egg extract, naturally arrested at metaphase of the second meiotic division (MII), lacks a functional SAC[51] excluding per se any SAC-dependent interference with APC/C activity. To obtain interphasic, unphosphorylated APC/C, which does not support CDC20-dependent activity, egg extract was treated with calcium to induce APC/C activation and, consequentially, CDK1 inactivation and entry into interphase[52,53]. Faster SDS-PAGE mobility of APC3 confirmed that the extract was interphasic after calcium treatment (Fig. 3E, samples 1 and 2). From interphasic extract, APC/C was immunoprecipitated using α-APC3 antibodies (sample 3) and phosphorylated on beads by recombinant PLK1 and CDK1/CKS1/cyclin B1$^{wt/mut}$ (CCC$^{wt/mut}$, sample 4). Kinase reactions were performed in the presence of the solvent control

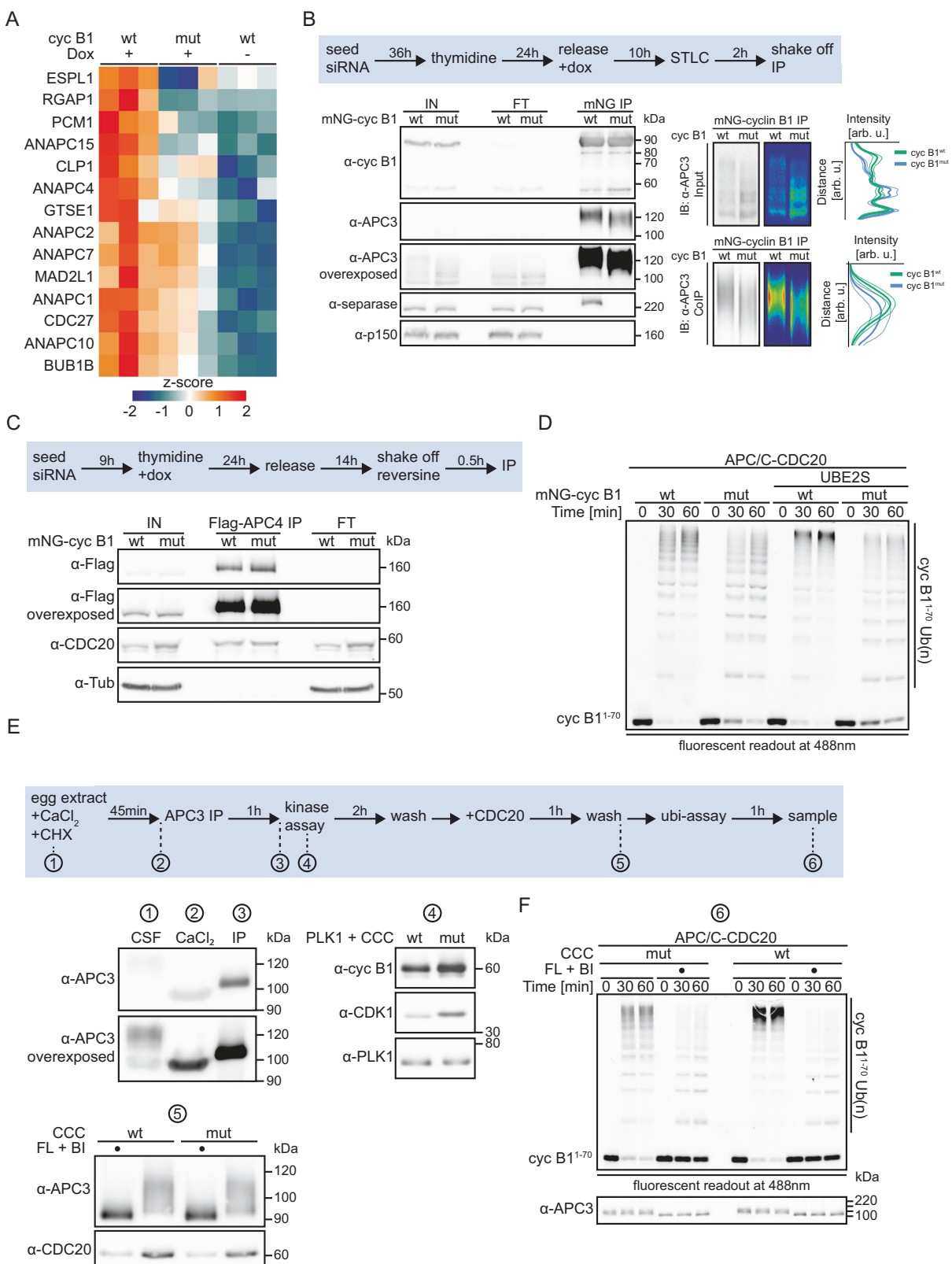

DMSO or the CDK and PLK1 inhibitors flavopiridol (FL) and BI2536 (BI), respectively. After washing, samples were incubated with recombinant CDC20 and washed again. Consistent with APC/C phosphorylation being critical for CDC20 recruitment[48–50,54–56] immunoblot of bead-bound APC/C revealed efficient CDC20 binding only when the APC/C was phosphorylated as indicated by slower SDS-PAGE mobility of APC3 (DMSO), but not in reactions containing flavopiridol and BI2536

(sample 5). Ultimately, bead-bound APC/C$^{CDC20}$ was used for in vitro ubiquitylation using recombinant ubiquitin, UBA1, UBE2C and cyc B1$^{1-70}$ (sample 6). As expected, only weak ubiquitylation activity was detected when flavopiridol and BI2536 were present during the in vitro phosphorylation reaction. Importantly, APC/C phosphorylated by CDK1/CKS1/cyclin B1$^{wt}$ showed a significantly stronger ubiquitylation activity compared to the one phosphorylated by CDK1/CKS1/cyclin

**Fig. 3 | Cyclin B1's phosphate-binding pocket is important for efficient APC/C activation. A** Heat-map visualizing the hierarchical clustering of the 14 significantly enriched proteins identified by quantitative DIA mass spectrometry in the Flag-cyclin B1$^{wt}$ over Flag-cyclin B1$^{mut}$ IP after ANOVA (s0 = 0.25, FDR = 0.05) and post-hoc Tukey HSD (FDR = 0.05). Technical replicates of n = 3 biological replicates are averaged and log2 abundances are z-score normalized. Euclidean distances are not shown, for details see methods. **B** Experimental outline of cell synchronization procedure. Immunoblot assessing α-mNG IP samples of mitotically arrested siB1&B2 HeLa cells expressing mNG-cyclin B1$^{wt/mut}$. p150 served as IP control. Right panels show the quantification of the APC3 signal intensity in the input and co-IP samples from n = 3 independent repetitions and plotted against the distance on the blot. **C** Schematic of experimental design of α-Flag-APC4 IP from mitotic siB1&B2 HeLa cells expressing mNG-cyclin B1$^{wt/mut}$ and treated for 30 min with 1 μM reversine for SAC silencing. Immunoblot assessing α-APC4 IP samples. **D** Fluorescent readout (488 nm) of in vitro ubiquitylation assay using immunoprecipitated APC/C from (**C**), recombinant ubiquitin, UBA1, UBE2C, where indicated UBE2S and as substrate fluorescein-labeled cyclin B1$^{1-70}$ (cyc B1$^{1-70}$). **E** Schematic of in vitro ubiquitylation assay using APC/C immunoprecipitated from interphasic Xenopus egg extract. To prevent cyclin B resynthesis following its calcium-induced degradation, extract was co-treated with the translation inhibitor cycloheximide (CHX). Phosphorylation by recombinant PLK1 and CDK1/CKS1/cyclin B1$^{wt/mut}$. In vitro kinase reactions were performed in the presence of DMSO or the CDK and PLK1 inhibitors flavopiridol (FL) and BI2536 (BI), respectively. Immunoblot assessing samples taken at indicated steps shown in experimental outline. **F** Fluorescent readout (488 nm) of in vitro ubiquitylation assay using APC/C purified according to (**E**) and supplemented with recombinant UBA1, UBE2C, ubiquitin and as substrate fluorescein-labeled cyclin B1$^{1-70}$. Immunoblot assessing phosphorylation state of APC3 under different experimental conditions.

B1$^{mut}$ (Fig. 3F). Again, this was particularly evident by the lack of highly ubiquitylated species.

Reportedly, CKS1 enhances CDK1 multisite phosphorylations by binding to primed substrates assisting downstream phosphorylation events even at non-consensus CDK1 sites[22–24,57]. As shown (Figs. 3B, S3B, and S3C), the APC/C (APC3, APC1) was hypophosphorylated in cyclin B1$^{mut}$ cells and displayed a slightly reduced binding to cyclin B1$^{mut}$ compared to cyclin B1$^{wt}$ cells. We, therefore, speculated that the APC/C is sequentially phosphorylated involving initial PBP-independent phosphorylation events that create phosphosite(s) that interact with cyclin B1's phosphate-binding pocket, thereby facilitating sequential PBP-dependent phosphorylation(s). In analogy to CKS1, we further speculated that the initial priming phosphosites of the APC/C preferentially match the CDK1 (S/T)P consensus motif, whereas downstream PBP-dependent phosphorylations not necessarily occur at consensus sites. Since the APC/C is phosphorylated at ~70 serine and threonine residues comprising consensus as well as non-consensus sites[48–50,54], a corollary of our hypothesis is that the APC/C should be less efficiently phosphorylated by CCC$^{mut}$ compared to CCC$^{wt}$, while a substrate containing a single consensus CDK1 site should be equally well phosphorylated by wt or mutant CCC.

To test this, we performed in vitro kinase assays using recombinant CCC$^{wt/mut}$ and apo-APC/C, both purified from insect cells, as well as MBP-Emi2$^{NT}$, which contains a single CDK1 consensus site. Of note, kinase assays were performed in the presence of both substrates strep-apo-APC/C and MBP-Emi2$^{NT}$. To focus exclusively on phosphorylations by CCC, PLK1 – in contrast to the assays shown in Fig. 3E – was omitted from the reaction. The phosphorylation pattern of the APC/C was evaluated by Phos-tag WB analyses using the antibodies available to us except for APC4 because it did not show any shift in its mobility (Fig. S3B). Since phosphorylation of Emi2$^{NT}$ did also not affect its mobility, we raised and validated a phospho-specific antibody (α-pT97) against the single CDK1 consensus site (Fig. S3E) so that we could monitor Emi2$^{NT}$ phosphorylation efficiency. APC3 – and to a lesser extent APC1 – were efficiently phosphorylated by CCC$^{wt}$ as indicated by a prominent shift in their mobility (Fig. 4A). In contrast, APC3 and APC1 were poorly phosphorylated by CCC$^{mut}$. Importantly, MBP-Emi2 was equally well phosphorylated by wt or mutant CCC as shown by α-pT97 WB analyses (Fig. 4A). Thus, these data support our hypothesis that the ability of cyclin B1's phosphate-binding pocket to bind pre-phosphorylated substrates is important for sequential APC/C phosphorylation events.

However, from these experiments, we were unable to determine whether in particular non-consensus CDK1 sites are phosphorylated in a pocket-dependent manner. To test this, we performed MS analysis of in vitro phosphorylated APC/C. Specifically, purified recombinant APC/C was treated with lambda (λ) phosphatase to obtain dephosphorylated APC/C, and subsequently re-phosphorylated using recombinant CCC$^{wt/mut}$ (Fig. 4B, S5 and Supplementary Data 1). Four independent APC/C purifications were analyzed in technical triplicates. After stringent data filtering for phosphosites and phosphopeptides (quantified in 3 out of 4 assays, present in > 50% of raw files, >95% of phospho localization probability; Fig. S4), we obtained 28 different phosphosites on 25 phosphopeptides on the APC/C incubated with CCC$^{wt}$. Of these sites, 18 phosphosites (64%) matched the minimal (S/T)P consensus motif of CDK1 and 10 (36%) were non-consensus sites (Fig. 4B). Using these high-confident quantified phosphosites as basis, APC/C samples phosphorylated by CCC$^{mut}$ were analyzed identically. We detected 11 phosphosites which passed the same filtering criteria with similar signal intensities in CCC$^{mut}$ (less than 2 fold change in intensity) and 17 sites which did not pass the filtering criteria because they were either not detected at all (12 phosphosites) or only rarely quantified (at least 2 fold reduced in intensity and in half or less than half of the raw files compared to wt; 5 phosphosites) compared to the wt (Fig. S4). Notably, all sites that were equally well quantified in wt and mutant CCC samples were exclusively consensus (S/T)P sites and all non-consensus sites were highly enriched or exclusively detected in the wt cyclin B1 samples, further strengthening our hypothesis that binding of cyclin B1's PBP facilitates primarily the phosphorylation of non-consensus CDK1 sites (Fig. 4B).

Previous studies on APC/C phosphoregulation revealed that its hyperphosphorylation by CDK1 is essential for binding of CDC20[48–50,59]. Specifically, an auto-inhibitory (AI) segment of APC1 acts as a molecular switch that in its unphosphorylated form obstructs engagement of CDC20 with the C-box binding site on APC8B and phosphorylation by CDK1/CKS1/cyclin B reliefs autoinhibition allowing CDC20 binding and APC/C activation. In our study, the non-consensus phosphosites enriched or exclusively detected in APC/C samples phosphorylated by CDK1/CKS1/cyclin B1$^{wt}$ are within the reported APC1 and APC3 loops, but also in APC5 and APC6 (Fig. 4B). To gain insights how these phosphosites might affect APC/C activity, we analyzed the association of binding partners and co-activator with the APC/C by MS analyses. APC/C complexes were APC3-immunopurified in independent triplicates from mitotically arrested siB1&B2 mNG-cyclin B1$^{wt/mut}$ HeLa cells. After filtering for quantifications in at least two per triplicate for each condition and imputation of missing values, we calculated the ratios between the measured log$_2$ intensities of the wt and mutant samples (log$_2$ intensity cyclinB1$^{wt}$ – log$_2$ cyclinB1$^{mut}$) of each APC/C subunit, CDC20, cyclin B1 and UBE2S. Then, data were normalized on the ratio of the bait protein APC3, and normalized fold changes of the proteins of interest are plotted (Fig. 4C). Ectopic cyclin B1 co-precipitated slightly more efficiently with the APC/C purified from cells expressing cyclin B1$^{wt}$ compared to cyclin B1$^{mut}$ expressing cells, confirming our cyclin B1 IP MS analyses (Fig. 3A). Notably, we observed no significant difference in the association of CDC20 or UBE2S with the APC/C purified from wt or mutant cyclin B1 expressing cells. Of note, we did not detect UBE2C in our MS analyses. Thus, pocket integrity while being critical for full APC/C phosphorylation and activity seems not to

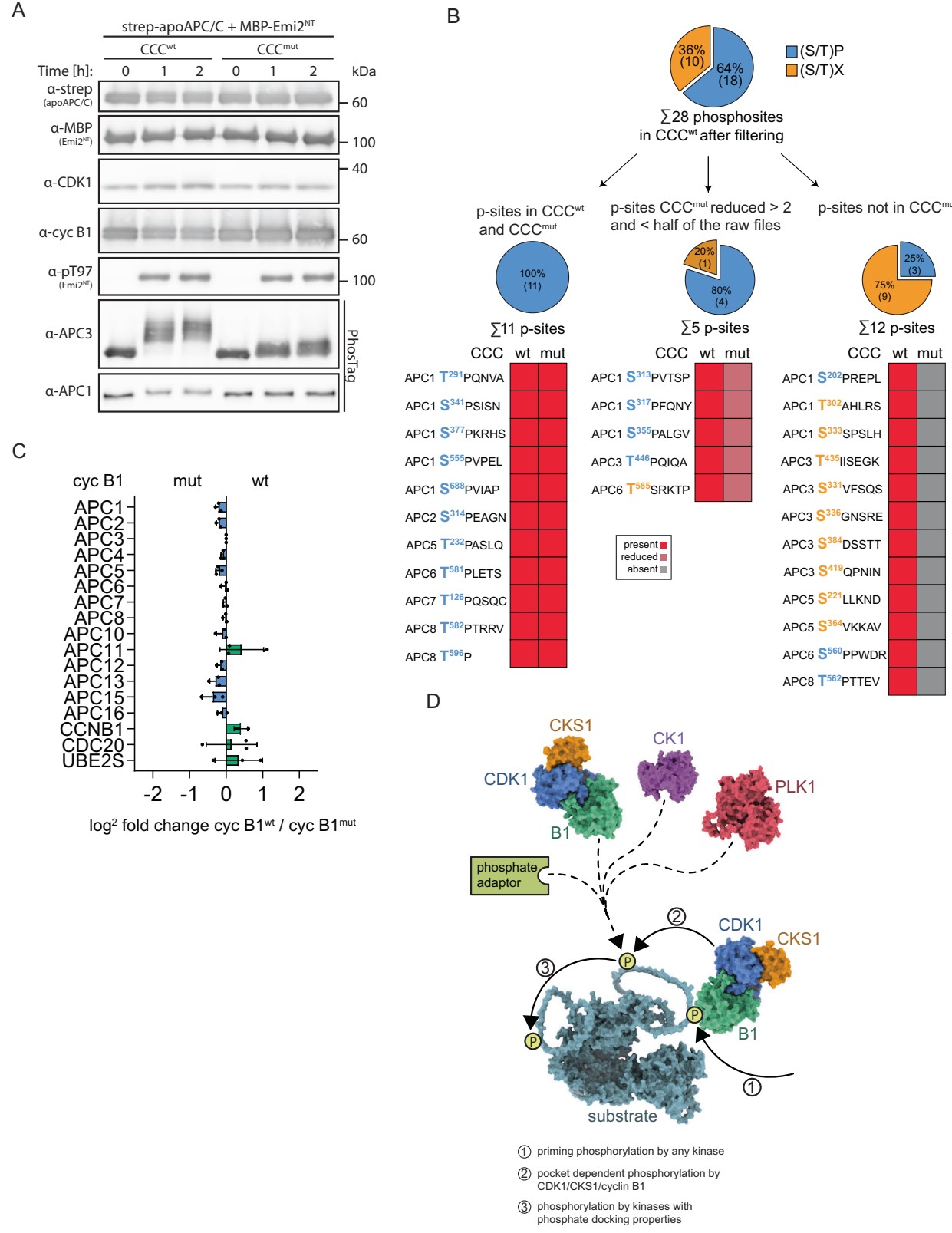

be involved in regulating the recruitment of the co-activator CDC20 or the E2 enzyme UBE2S.

## Discussion

Phosphorylation of the plethora of CDK/cyclin substrates must occur with high specificity and in the correct temporal order to ensure accurate cell cycle progression. We demonstrate here that cyclin B1's phosphate-binding pocket is a major determinant for faithful execution of mitosis adding a layer of complexity in regulated substrate phosphorylation by CDK1/cyclin B1. The starting point for this study was the recently identified positively charged pocket in cyclin B1, which engages separase phosphoserine-1126 and thereby enables

**Fig. 4 | Cyclin B1's phosphate-binding pocket is important for APC/C phosphorylation at non-consensus CDK1 sites. A** In vitro kinase assay using recombinant CCC[wt/mut] to phosphorylate recombinant strep-tagged apo-APC/C and MBP-Emi2[NT], both present in the same kinase reaction mixture. Shown are western blots for all relevant proteins involved in the reaction. As readout for phosphorylation efficiency, pT97 (MBP-Emi2[NT]) and APC3 (apoAPC/C) phosphorylation pattern were used. APC3 and APC1 blots were conducted on PhosTag™ gels. In vitro kinase reactions were performed at 30 °C, and at indicated timepoints samples were taken and analyzed by WB. **B** Pie charts and discrete heatmaps of APC/C phosphosites identified after in vitro phosphorylation by CCC[wt/mut]. Dephosphorylated and purified APC/C has been subjected to an in vitro phosphorylation assay using either cyclin B1[wt] or cyclin B1[mut] and phosphosites have been identified by MS. The phosphosites were stringently filtered (see Fig. S4 for details). Then, the obtained phosphosites from CCC[wt] were divided in either minimal CDK1 (S/T)P consensus sites (blue) or non-consensus sites (orange). The upper pie chart shows the percentage and number of identified consensus or other phosphosites in the CCC[wt] samples. Using these phosphosites as basis, the CCC[mut] data set was analyzed and classified in sites that were equally well detected in CCC[mut] samples (left) and sites that were either rarely (at least 2 fold reduced in intensity and in half or less than half of the raw files compared to wt; middle) or not at all detected (right). Below the

pie charts heatmaps of the identified phosphosites are shown (red: present; light red: reduced at least 2 fold in half or less than half of the rawfiles compared to CCC[wt]; gray: absent. **C** Bar plot of the fold changes of APC/C subunits, CDC20, cyclin B1 and UBE2S of the immunopurified APC/C of cells overexpressing either cyclin B1[wt] or B1[mut]. APC3 was immunopurified from siB1&B2 cells expressing either cyclin B1[wt] or B1[mut] and interaction partners were quantified by MS. For proteins of interest, the ratio of the quantified protein intensities between cyclin B1[wt] and cyclin B1[mut] was calculated (log₂ intensities cyclin B1[wt] – log₂ intensities B1[mut]) and normalized on the ratio of the bait protein APC3. The x-axis shows the normalized log₂ changes of relevant indicated proteins (y-axis). The experiment was conducted in n = 3 independent biological replicates, each dot represents a single measurement. Data are represented as mean values ±SD. **D** Model of pocket-dependent multisite substrate phosphorylations. Step 1: Priming phosphorylation carried out by any kinase creates a docking site for cyclin B1's phosphate-binding pocket. Step 2: Upon pocket-dependent recruitment, CCC catalyzes the phosphorylation of an additional site. Step 3: This phosphosite can serve as docking site for CCC itself using cyclin B1 or CKS1 as phosphate adaptor or other kinases with phosphate adaptor properties such as casein kinase 1 (CK1) or PLK1. For illustration, CCC (PDB: 7NJ0) is shown in blue (CDK1), orange (CKS1), and green (cyclin B1), PLK1 (AlphaFold2 prediction) in red, and CK1 (PDB: 6GZD) in purple.

complex formation critical for mutual inhibition of CDK1/cyclin B1 and separase[28,29]. Within this pocket, residues critical for recognition of the phosphate group are present in B-type, but not A-, D-, or E-type cyclins. Mapping the sequence conservation onto the structure of cyclin B1, using a multiple sequence alignment that includes more than 2000 sequences, reveals that cyclin B1's PBP is highly conserved during evolution (Fig. 1B). Despite the high degree of sequence conservation across species including yeast, the mechanism of mutual inhibition of separase and CDK1/cyclin B is restricted to vertebrates[60], suggesting that the pocket evolved originally to fulfill separase-independent functions. Indeed, for yeast it was recently reported that multisite phosphorylation of the transcriptional co-activator Ndd1 by CDK1 seems to involve both CKS1 and Clb2's phosphate-binding pocket[32].

By combining cellular studies, MS-based proteomics and phosphosite analysis as well as in vitro reconstitution, we discovered that mitotic fidelity in human cells critically depends on the integrity of cyclin B1's phosphate-binding pocket. Cells expressing mutant cyclin B1 in the background of depleted endogenous cyclin B1 and cyclin B2 display multiple mitotic defects causing prolonged SAC activation (Figs. 2C and S2G). Notably, cells expressing pocket mutant cyclin B1 display a higher frequency of lagging chromosomes upon anaphase onset compared to wt cyclin B1 expressing cells (Fig. 2G) suggesting that pocket integrity is essential for mitotic fidelity. In search for the mechanisms underlying the multiple mitotic defects, we first investigated the localization of pocket mutant cyclin B1 and observed that it is unable to bind to kinetochores but still localizes to spindle poles, which is in line with previous reports showing that the hydrophobic patch within the cyclin box mediates recruitment of cyclin B to centrosomes[41,61]. Mad1 targets cyclin B1 to kinetochores[43] and we observed that Mad1 binds slightly less efficient to mutant cyclin B1, compared to wildtype cyclin B1 (Fig. S2E). Our immunofluorescence analyses suggest that pocket integrity is important for Mad1 localization to the kinetochore corona, but not to the outer kinetochore (Fig. S2F). Yet, further studies are required to dissect in detail the molecular mechanism underlying cyclin B1's kinetochore localization and its contribution to SAC signaling. As a complementary approach towards understanding the molecular mechanisms underlying the observed mitotic defects, we performed MS analyses and identified *inter alia* PCM1 and GTSE1 as potential pocket dependent interactors of cyclin B1 (Fig. 3A). PCM1 is a component of the centriolar satellites relevant for centrosome assembly and function and was shown to be important for the proper recruitment of several centrosomal proteins[62–64]. CDK1-mediated phosphorylation of the PCM1-localized

protein cenexin results in recruitment of PLK1 and this is important for proper function of PCM proteins[65]. Based on our data, it is conceivable that cyclin B1[mut], while being able to localize to centrosomes, is unable to catalyze pocket-dependent phosphorylations important for this pathway. GTSE1 binds to plus-ends of microtubules in an EB1-dependent manner and modulates microtubule dynamics[66]. Phosphorylation of GTSE1 by CDK1/cyclin B in early mitosis abolishes its interaction with EB1 resulting in the destabilization of astral microtubules facilitating cells to reorient their spindles[66]. Thus, it warrants further investigations if GTSE1 phosphorylation and, thus, regulation of microtubule dynamics relies on pocket integrity. Of note, kinetochore-microtubule attachments are highly dynamic in prometaphase, which is important for efficient correction of attachment errors, and the switch to more stable attachments in metaphase depends on proteasomal degradation of cyclin A[67]. Since cyclin A lacks a functional pocket (Fig. 1B), it is tempting to speculate that pocket-dependent phosphorylations mediated by CDK1/cyclin B contribute to adjustment of microtubule dynamics as cells progress through mitosis and this function is important for timely satisfaction of the SAC. In addition, we observed a SAC-independent delay in anaphase onset in pocket mutant expressing cells (Figs. 2C, D, and S2C). MS analyses revealed enrichment of several APC/C subunits in wt cyclin B1 IP samples compared to mutant samples (Fig. 3A). Activity of mitotic APC/C is tightly controlled by the SAC as well as by posttranslational modifications to ensure that its substrates are targeted for degradation on time[54–56,58,68–71]. Our MS-based analyses identified phosphosites that were equally well phosphorylated by wt and mutant CDK1/CKS1/cyclin B1 (Fig. 4B). Notably, all of these sites match the minimal CDK1 consensus motif (S/T)P. In contrast, all non-consensus sites were highly enriched or exclusively detected in wt CDK1/CKS1/cyclin B1 samples. Thus, in line with our starting hypothesis, these data suggest that cyclin B1's pocket is important for the phosphorylation of low-affinity sites such as non-consensus CDK1 sites via docking to primed substrates. Sites that were enriched or exclusively detected in CCC[wt] samples also include (S/T)P sites (Fig. 4B), suggesting that they might also be low-affinity phosphorylation sites. Consistent with the requirement of pocket integrity for high occupancy of APC/C phosphorylation, APC1 and APC3 phosphorylated by pocket mutant CCC display faster SDS-PAGE mobility (Figs. 3B and 4A). In vitro ubiquitylation assays revealed that APC/C purified from pocket mutant expressing cells (Fig. 3D) or in vitro phosphorylated by CDK1/CKS1/cyclin B1[mut] (Fig. 3F) is less efficient in forming highly ubiquitylated substrate species. Currently, we have no evidence if compromised activity in forming poly-ubiquitin chains or in mono-ubiquitylation

at multiple sites accounts for the observed defect in forming highly ubiquitylated substrate species. Reportedly, phosphorylation-dependent recruitment of CDC20 to the APC/C is a major regulatory mechanism of APC/C activity[48–50]. Specifically, a phosphorylation cascade that involves recruitment of the CCC complex to a hyperphosphorylated loop in APC3 results ultimately in phosphorylation of an auto-inhibitory segment in APC1 (APC1$^{L300}$, aa 307-395) by CDK1/CKS1/cyclin B. Upon hyperphosphorylation, this segment dissociates from the C-box binding site in APC8B, thereby allowing CDC20 to bind and activate the APC/C[48,49,72]. Complementary studies using APC/C purified from Xenopus egg extract and in vitro phosphorylated by CDK1/CKS1/cyclin B[72] and human recombinant APC/C in vitro phosphorylated by CDK2/CKS2/cyclin A3 and PLK1[48], identified different phossites that were critical for CDC20 recruitment. Only one phosphosite, pS377, was identified in both studies. Thus, in line with another study on APC/C phoshoregulation[49], these data suggest that there is a certain degree of redundancy with respect to the individual sites that can be phosphorylated in order to allow CDC20-mediated APC/C activity. In our study, we did not observe significant differences in the recruitment of CDC20 to the APC/C phosphorylated by wt or mutant cyclin B1 (Fig. 3C, E) implicating that the phosphosites that were equally well detected in wt and mutant CCC samples (Fig. 4B) are sufficient for CDC20 recruitment. Consequentially, this observation suggests that pocket-dependent APC/C phosphorylation regulates APC/C activity by a mechanism distinct to CDC20 binding. Reportedly, cyclin B1 and the APC/C are both recruited to nucleosomes via an "arginine-anchor", and this interaction is important for efficient cyclin B1 degradation[73]. Although we cannot completely rule out the possibility that cyclin B1's pocket acts in this pathway, it is unlikely to be responsible for the reduced APC/C activity we observed with pocket mutant cyclin B1. This conclusion is supported by two findings: (i) reduced APC/C activity was identified in a reconstituted in vitro ubiquitylation assay lacking chromatin (Fig. 3D, F), and (ii) the degradation of securin, which does not possess an arginine-anchor, was also impaired in cells expressing pocket mutant cyclin B1 (Fig. S3A). We instead prefer the possibility that pocket-dependent phosphorylations stabilize the active APC/C$^{Cdc20}$-substrate complex in a favorable conformation that allows enhanced substrate ubiquitylation. Such a productive conformation could influence CDC20's binding to the APC/C or enhance substrate recognition. One function of the co-activator CDC20 is to provide degron recognition sites present in most APC/C substrates. Another function of the co-activator is to stimulate APC/C activity by inducing a conformational change of the catalytic module from a "down" to an "up" conformational state. This includes both possibilities, the attachment of multiple single ubiquitin molecules to the substrate mediated by UBE2C, as well as ubiquitin chain formation carried out by UBE2S[74,75].

Based on our data, we propose a model according to which multisite substrate phosphorylations are initiated by a priming phosphorylation event catalyzed by CDK1 or any other mitotic kinase (Fig. 4D, step1). The created phosphosite serves as docking site for the PBP of B-type cyclins facilitating subsequent phosphorylation (step 2). Depending on the substrate, this phosphosite could support further phosphorylation events by recruiting a phosphosite adaptor such as B-type cyclins, CKS proteins, PLK1, or casein kinase 1 (CK1). We further speculate that B-type cyclins, by engaging CDK1 to primed substrates, enable CDK1 to phosphorylate non-consensus sites. Thus, the phosphate-binding pocket of B-type cyclins could act as a specificity determinant that by enhancing the phosphorylation of non-consensus sites by CDK1 distinguishes CDK1 from other proline-directed kinases such as MAPK, which have no phosphosite docking capability, and therefore are unlikely to phosphorylate non-consensus sites. Notably, non-proline directed CDK1 phosphorylation has been reported previously[24,76–79] and, in fact, a recent study revealed that 70% of cell cycle regulated phosphorylations for which the responsible kinase was unknown turned out to be non-proline directed CDK1 sites[24]. In this case, cyclin A and CKS1 enhance phosphorylation at non-consensus sites with cyclin A shifting substrate specificity of CDK1 and CKS1 enhancing CDK1-primed multisite phosphorylation. Notably, CKS proteins display a strict phosphothreonine preference[57] limiting their versatility. In contrast, our study revealed that cyclin B1's pocket accommodates not only separase pS1126, but also threonine when phosphorylated or a phosphate-mimicking S1126E mutation (Fig. S1F). Thus, acidic residues might circumvent the need of priming phosphorylations for subsequent pocket-dependent phosphorylation. Such a mechanism would amplify the level of complexity of CDK1/cyclin B phosphorylations and, therefore, warrants further investigation.

## Methods

### Cell culture

HeLa FRT/TO cells with endogenously tagged APC4 (3xFlag-mVenus-APC4) were a gift from J Mansfeld (Institute of Cancer Research, London[80]). Mammalian tissue culture cells were cultured according to standard cell culture methods. HeLa cells were cultured and passaged in DMEM (Gibco) supplemented with 9% FBS (Gibco) and 0.1 mg/ml normocin (Invivogen). Stably integrated cells were selected with 0.4 mg/ml hygromycin (Invivogen). Experiments were performed without antibiotics. Cells were screened for mycoplasma contamination regularly.

For cell extracts, HeLa cells were harvested and washed two times with 1X PBS. Cells were centrifuged at $1500 \times g$ for 1 min and supernatant was removed and the cell pellet was frozen in liquid nitrogen. Thawed cell pellets were resuspended with NETN buffer (25 mM Tris-Cl pH7.8, 150 mM NaCl, 1 mM EDTA, 0.1 % NP-40, 5 mM NaF, 10 mM β-glycerophosphate, 1 mM DTT, 1x cOmplete protease inhibitor tablet [Roche], 1x PhosStop [Roche]). The mixture was incubated on ice for 15 min. Afterwards, the lysate was cleared by centrifugation at $1500 \times g$ for 10 min.

### Cell cycle synchronization

For single thymidine block, cells were treated with thymidine (2 mM) for 24 h before releasing into fresh medium. For double thymidine block, cells were treated with thymidine (2 mM) for 20 h before releasing into fresh medium for 10 h. Afterwards, cells were treated again with thymidine (2 mM) for 18 h, before releasing again into fresh medium. For SAC inhibition, reversine (1 μM) was added 6 h after the last thymidine release. For APC/C immunoprecipitation cells were treated with reversine (1 μM) for 0.5 h before harvesting to reduce associated MCC components. For prometaphase arrest, cells were treated with nocodazole (333 nM) for 20 h and harvested by mitotic shake off.

### Generation of mNG-/3xFlag-cyclin B1$^{wt/mut}$ HeLa cells

For the generation of stable cell lines, HeLa FRT/TO cells with randomly integrated H2B-mCherry and an empty FRT-site were transfected using X-tremeGENE HP (Sigma Aldrich) transfection reagent according to the manufacturers protocol. Therefore, pCDNA5-FRT-TO vector with the inserted transgene was co-transfected with pOG44 (Flp recombinase) in a molar ratio of 1:10. Afterwards the cells were incubated for 3 days in medium without antibiotics. Selection for correctly inserted clones was pressured by the addition of 0.4 mg/mL hygromycin to the medium.

### siRNA and DNA transfection

For RNAi mediated knockdown of proteins 20–40 nM of siRNA (Dharmacon Horizon) were mixed with RNAiMAX lipofectamine (Invitrogen) and Opti-MEM reduced serum medium (Gibco) according to the manufacturer's protocol. Cells were analyzed for knockdown efficiency after 24–48 h post transfection. siRNAs were purchased

from Dharmacon Horizon: cyclin B1: 5'-CAACAUUACCUGUCAUAUA-3'; cyclin B2: 5'-GUACAUGUGCGUUGGCAUU-3'.

## Immunofluorescence

Cells were seeded on precoated coverslips (13 mm diameter, thickness no. 1.5, VWR). For fixation, cells were treated with 3.7% formaldehyde solution in 1X PERM buffer (100 mM Pipes pH 6.8, 10 mM EGTA, 1 mM MgCl2, 0.2% Triton X-100) for 20 min at RT. Afterwards cells were washed with 1X TBSTx (Tris buffered saline 0.15 M NaCl, 0.02 M Tris-Cl pH 7.4, 0.1% Triton X-100). Cells were incubated in Abdil (1X TBSTx, 1% BSA) overnight, 4 °C. Next, the coverslips were washed two times with 1X TBSTx. The primary antibodies were diluted in Abdil and incubated for 2 h at RT. Coverslips were washed two times with 1X TBSTx. The secondary antibodies were diluted in Abdil and applied to the coverslips for 3 h at RT. Cells were again washed two times with 1 x TBSTx before applying a Hoechst (Thermo Fisher) solution in 1X PBS for 10 min at RT. Afterwards, the cells were washed three times with 1X PBS before mounting (ProLong Diamond antifade mountant, Thermo Fisher) onto glass slides (Superfrost, VWR). The mountant was dried overnight at 4 °C. The cells were imaged using a DeltaVsision Core system (GE Healthcare) mounted on an IX-71 inverted microscope (Olympus) equipped with an environmental chamber (Applied precision) and a CoolSnap HQ$^2$ camera (Photometrics). Images were acquired using a 60x/1.40 NA UApo (Olympus) oil objective or a 100x/1.40 NA UPLS Apo (Olympus) oil objective. Images were acquired and processed using SoftWorx software. Or a LSM880 with AiryScan (Zeiss) with a 63x/1.40 NA Plan-Apo (Zeiss) oil objective. Images were acquired and processed using Zeiss ZEN-Black software.

## Live cell imaging

Cells were seeded in 12- or 24-well glass-bottom plates (IBL/ZellKontakt). Expression of exogenous integrated constructs was induced by the addition of doxycycline (150–600 pg/mL) at least 24 h before imaging. Cells were imaged using the CellDiscoverer 7 (Zeiss) or the Axio Observer Z1 (Zeiss) equipped with an environmental chamber (Zeiss), Colibri LED module and CoolSnap-ES2 camera (Photometrics). Cells were incubated at 37 °C and 5% CO$_2$ for up to 24 h. Imaging with the CellDiscoverer 7 required the use of DMEM (Gibco) without phenol red supplemented with 9% FBS and 1% (v/v) Glutamax. Imaging with the Axio Observer Z1 required the use of CO$_2$-independent media (Gibco) supplemented with 9% FBS and 1% (v/v) Glutamax. Images were taken every 2–5 min with a 20x/0.95 NA Plan-Apo autocorr. (Zeiss) air objective or 50x/1.20 NA Plan-Apo autocorr. (Zeiss) water immersion objective. Images were acquired and processed using MetaMorph, VisiView, Zeiss ZEN-Blue or ImageJ software.

## In vitro kinase assay

Recombinant $^{MBP}$CDK1$^{StrepII}$/cyclin B1$^{wt/mut\ 8xHis}$ (1.4 ng/μL) were mixed with substrate (0.09 mg/mL) in freshly prepared kinase assay buffer (12 mM HEPES pH 8.0 (NaOH), 85 mM NaCl, 20 mM MgCl$_2$, 90 mM β-glycerophosphate, 23 mM EGTA, 2 mM DTT, 1x cOmplete protease inhibitor [Roche], 1x PhosStop [Roche], 1% glycerol, 1 mM ATP and 75 nM P33-ATP). The reaction was incubated in a thermomixer at 30 °C, 1500xg. Samples were taken at indicated timepoints and mixed 1:1 with 3X Laemmli buffer.

## Immunoprecipitation

For bead preparation, 100 μl magnetic Protein G Dyna Beads (Invitrogen) were coupled with 10 μg APC3 or Flag antibody (ab). For APC/C immunoprecipitation (IP) from HeLa cell extract, APC3 or Flag antibody coupled beads were mixed in a 1:2 ratio with HeLa cell extract and incubated for 1.5 h at 4 °C. Afterwards, beads were washed two times with 1X PBST (0.025 % Tween20) and two times with 1X PBS. Beads were then either used for ubiquitylation assays, eluted with 1X Laemmli buffer for WB or eluted with 6 M urea for mass spectrometry purposes.

For APC/C IP from interphasic CSF extract, APC3 antibody coupled beads were mixed in a 1:2 ratio with interphasic CSF extract and incubated for 1 h at RT on a rotator. Afterwards, beads were washed three times with 1X PBS. mNeonGreen-trap magnetic agarose beads (Chromotek) were used for immunoprecipitation of mNG-cyclin B1 from HeLa cell lysates according to the manufacturers protocol.

## In-vitro phosphorylation assay of interphasic *X. laevis* APC/C

Immunoprecipitated interphasic *X. laevis* APC/C was used as a substrate to be phosphorylated by 0.14 mg/mL $^{8xHis}$CDK1/CKS1/cyclin B1$^{wt/mut\ StrepII}$ and 0.05 mg/mL PLK1 in kinase assay buffer (12 mM HEPES pH8.0 (NaOH), 85 mM NaCl, 20 mM MgCl$_2$, 90 mM β-glycerophosphate, 23 mM EGTA, 2 mM DTT, 1x cOmplete protease inhibitor[Roche], 1x PhosStop [Roche], 1% glycerin, 1 mM ATP) The reaction was incubated for 2.5 h at 30 °C and 1500 × g. Afterwards, the beads were washed three times with 1X PBS to remove residual kinases. Recombinant CDC20 (0,136 mg/mL) was mixed with kinase assay buffer (without ATP) and incubated with the phosphorylated APC/C for 1 h at 30 °C and 1500 × g. Afterwards, beads were washed three times with 1X PBS.

## In vitro ubiquitylation assay

Either in-vitro phosphorylated APC/C (*X. laevis*) or immunprecipitated APC/C (HeLa) were mixed with recombinantly produced Uba1 (0.15 μM), Ube2C (2.5 μM), Ubiquitin (13 μM) and cyclin B1 N-term (1–70) (0.85 μM) plus Ubiquitin-aldehyde (5 μM) (Biomol) in ubi-assay buffer (20 mM HEPES pH7,5 (KOH), 100 mM KCl, 3 mM MgCl$_2$, 1 mM CaCl$_2$, 1 mM DTT, 1% glycerol, 15 μM MG132, 2.5 mM ATP). The Reaction was incubated at 30 °C and 1500 × g. Samples were taken at indicated timepoint and mixed 1:1 with 3X Laemmli Buffer.

## Expression and purification of APC/C

APC/C expressing baculoviruses were kindly donated from David Barford lab[81]. APC/C is expressed in SF9 cells by co-infection with two baculoviruses encoding the subunits of APC/C at a cell concentration of $1.5 \times 10^6$ cells/mL. Cells are harvested 48–72 h post-transfection. APC/C purification is performed based on previous purification protocols, introducing experiment-specific protocol modifications[48]. Cell pellets are resuspended in lysis buffer (250 mM NaCl, 50 mM Tris pH 8.3, 5% glycerol, 2 mM DTT, protease inhibitor cocktail tablets (cOmplete, Roche) and 5 units/mL Supernuclease (Novagen)) and lysed through sonication. The lysate is centrifuged for 1 h at 32,000 × g and applied to StrepTactin Superflow Cartridge (Qiagen). The column is washed with wash buffer (250 mM NaCl, 50 mM HEPES pH 8, 2 mM DTT) until the UV absorption stabilizes. APC/C is eluted with wash buffer supplemented with 2.5 mM desthiobiotin. Peak fractions are collected and dephosphorylated overnight at 4 °C with lambda phosphatase (1:10 molar ratio – APC/C: lambda phosphatase). Additional 0.5 mM MnCl$_2$ are added to the reaction buffer for dephosphorylation. Dephosphorylated APC/C is diluted to a final buffer concentration of 125 mM NaCl, 50 mM HEPES pH 8, 1 mM DTT and applied to a HiTrap Q HP column (Cytiva). Elution is performed by applying a salt gradient ranging from 125 mM NaCl to 1 M NaCl. Dephosphorylated APC/C is collected, concentrated, and injected onto a Superose 6 Increase 10/300 GL column (Cytiva), equilibrated in size-exclusion buffer (200 mM NaCl, 20 mM HEPES pH 8).

## Purification of CCC and CCC mutants

CCC$^{wt}$ and CCC$^{mut}$ were purified as previously published[29].

## Expression and purification of lambda phosphatase

Lambda phosphatase is expressed in *E. coli* BL21 cells at 37 °C. Cells are grown to an OD of 0.6 and induced with 0.05 mM IPTG. The temperature was lowered to 20 °C and the cells were harvested after overnight incubation. Cells were resuspended in lysis buffer

containing 500 mM NaCl, 50 mM Tris-HCl pH 7.5, 2 mM EDTA and 10% glycerol (v/v). After cell lysis and centrifugation, lambda phosphatase is purified through $Ni^{2+}$-affinity purification. The remaining imidazole is removed through over-night dialysis, followed by size-exclusion chromatography with a final buffer composition of 250 mM NaCl, 25 mM Tris-HCl pH 7.5, 10% glycerol (v/v) and 0.5 mM $MnCl_2$.

### In vitro phosphorylation assay of recombinant APC/C for Phos-Tag SDS-PAGE analyses

Recombinant apoAPC/C (0,2 mg/ml) and MBP-Emi2$^{NT}$ (0,05 mg/ml) were phosphorylated in vitro using either CCC$^{wt}$ or CCC$^{mut}$ (0,1 mg/ml) for 2 h at 30 °C in the reaction Buffer (12 mM HEPES pH8.0, 85 mM NaCl, 20 mM $MgCl_2$, 90 mM β-Glycerophosphate, 23 mM EGTA, 2 mM DTT, 1x cOmplete, 1x PhosStop, 1 % Glycerin, 1 mM ATP). Samples are loaded onto SDS-PAGE Gels with or without the addition of 5 μM PhosTag™.

### In vitro phosphorylation assay of recombinant APC/C for MS analyses

Dephosphorylated APC/C is incubated with CCC wt or mut (10:1 molar ratio) in reaction buffer (150 mM NaCl, 50 mM Hepes pH 8, 10 mM $MgCl_2$, 5 mM ATP, 1 mM orthovanadate (Thermo Scientific)). The reaction is incubated for 3 h at RT. Afterwards, it is applied to an analytical gel filtration to remove CCC from the final sample (Superose 6 Increase 5/150 GL, equilibrated in 150 mM NaCl, 50 mM Hepes pH 8). The peak fraction is collected, concentrated, and denatured with 10 M urea to a final protein concentration of 0.4–0.6 mg/mL and a urea concentration between 6 M and 7.5 M. Afterwards, samples are analysed by mass spectrometry.

### Protein purification

The expression and purification of human separase-securin fusion constructs containing C2029S/S1126T or C2029S/S1126E mutations, as well as the CDK1/cyclin B1/CKS1 complex, was previously described[29]. Cyclin B1 fragment (158-433 aa) was cloned into pETM41 vector harboring an N-terminal 6xHis-MBP tag for BL21 (DE3) *Escherichia coli* expression. Six liters of LB media were inoculated with overnight cultures and incubated for around 4 h at 37 °C when the OD$_{600nm}$ reached 0.6. Temperature was decreased to 18 °C and 0.5 mM isopropyl β-D-1-thiogalactopyranoside (IPTG) was added for overnight expression. Wild type cyclin B1$^{158-433}$ or cyclin B1$^{158-433}$ mutant (Arg307, His320 and Lys324 were mutated to Glu307, Phe320 and Glu324 respectively) was purified using a His-trap HP column in 50 mM HEPES pH 8.0, 800 mM NaCl, 20 mM imidazole, 10 mM β-mercaptoethanol and 10% glycerol lysis buffer and subsequently eluted by an imidazole gradient from 20 mM to 500 mM. The elution containing cyclin B1 protein was cleaved by TEV protease to remove His-MBP tag during dialysis against 50 mM HEPES pH 8.0, 800 mM NaCl, 5 mM β-mercaptoethanol and 10 % glycerol. Cleaved cyclin B1 was concentrated and further purified by Superdex 200 Increase 10/300 GL gel filtration column (20 mM HEPES pH 8.0, 600 mM NaCl, 5% glycerol and 2 mM DTT). Monodisperse peak of cyclin B1 was collected, flash-frozen in liquid nitrogen and stored at −80 °C.

CKS1 was cloned into pETM41 vector and purified with similar procedures as cyclin B1 except the last step. Briefly, cleaved CKS1 was concentrated and loaded onto Superdex 75 Increase 10/300 GL gel filtration column equilibrated with 20 mM HEPES pH 8.0, 150 mM NaCl, and 2 mM DTT.

For CDK1$^{MBP}$-cyclin B1 constructs, CDK1 and cyclin B1 were cloned into modified pF1 vector for co-expression in insect cells with CDK1 containing an N-terminal His-MBP tag and cyclin B1 containing a C-terminal Twin-StrepII tag. Typically, 20 ml recombinant P3 baculovirus was used to infect 500 ml of Sf9 insect cells (Invitrogen) at a cell density of roughly $2.0 \times 10^6$ cells/ml. The cells were incubated for 48 h at 27 °C at 100 rev/min, harvested at a cell viability rate of 80–90%,

flash-frozen in liquid nitrogen and stored at −80 °C. Cells expressing CDK1$^{MBP}$-cyclin B1 were resuspended in lysis buffer containing 50 mM HEPES pH 8.0, 500 mM NaCl, 10 mM β-mercaptoethanol, protease inhibitor cocktail and 5% glycerol, sonicated for 5 min and centrifuged at $25,000 \times g$ for 1 h. The supernatant was filtered using 0.45 μm filters and applied to a 5 ml StrepTactin Superflow Cartridge (Qiagen). The column was extensively washed with lysis buffer without protease inhibitor and the protein was eluted by adding 2.5 mM desthiobiotin. Elution containing CDK1$^{MBP}$-cyclin B1 was further purified by a Histrap HP column and a final size-exclusion step using a Superdex 200 Increase 10/300 GL column in 20 mM HEPES pH 8.0 150 mM NaCl and 0.5 mM TCEP was performed.

For purification of Emi2, the fragment was cloned into a pMAL-vector with N-terminal MBP-tag and C-term His-tag. Expression took place in BL21 DE3 RIL induced with 500 μM IPTG for 3–4 h at 37 °C. Bacteria were centrifuged for 10 min at $3000 \times g$ and the pellet was resuspended in 20 mL lysis buffer (20 mM Tris-HCl pH 8.0, 300 mM NaCl, 5 Mm Imidazole and 1x Protease Inhibitor). After snap freeze in liquid $N_2$ thawed bacteria pellet solution was lysed with the Emulsi-Flex-C3. Lysate was cleared by centrifugation at $20,000 \times g$, 4 °C for 30 min. The supernatant was mixed with 1 mL Ni-NTA beads (Qiagen) and washed with 10 mL PBS before use. Bead and lysate were incubated for 2 h at 4 °C rotating. Beads were washed three times with washing buffer (20 mM Tris HCl pH 8, 300 mM NaCl, 20 mM Imidazole, 0.1% Triton-X-100) containing 1 mM ATP and 1 mM $MgCl_2$ and three times with washing buffer. Elution took place with elution buffer (20 mM Tris HCl pH 8.0, 300 mM NaCl and 200 mM Imidazole). Peak fractions were identified via Coomassie, pooled and dialyzed with PBS with 10% Glycerol overnight at 4 °C. For storage dialyzed protein was aliquoted, snap frozen and kept at −80 °C.

For expression of Plk1 WT or KD, Sf9 cells were infected with P1 and incubated for 48 h and 3 h before harvesting cells were treated with 100 nM okadaic acid. Cells were harvest by centrifugation at $1000 \times g$ for 5 min. Cell pellet were flash frozen until purification. For purification cell pellets were resuspended in 1 mL lysis buffer (10 mM HEPES pH 7.5, 20 mM β-Glycerophosphate, 1 mM EGTA, 5 mM β-Mercaptoethanol, 0.1 mM Na-vanadate, 2x Protease-Inhibitor cOmplete (Roche), 150 mM NaCl, 1% CHAPS, 20 mM Imidazole) per $2 \times 10^7$ cells. Cell lysate was cleared by centrifugation at 4 °C $25,000 \times g$ for 30 min followed by filtration (0.45 μm). Lysate was incubated for 2 h with Ni-NTA beads (Qiagen) previously washed with lysis buffer. Beads were washed with around 50 to 75 column volumes washing buffer (10 mM HEPES pH 7.5, 0.1 mM Na-vanadate, 20 mM β-glycerophosphate, 1 mM EGTA, 5 mM β-Mercaptoethanol, 500 mM NaCl, 0.1% CHAPS, 1x Protease-Inhibitor cOmplete (Roche), 20 mM Imidazole). Protein was eluted in 10 fractions with elution buffer (10 mM HEPES pH 7.5, 25 mM NaCl, 2 mM $MgCl_2$, 1 mM EGTA, 50 mM sucrose, 5 mM β-mercaptoethanol, 500 mM Imidazole). Fractions with similar protein amount were pooled and dialyzed against 10 mM HEPES pH 7.5, 150 mM NaCl, 1 mM DTT, 200 mM Imidazole, 10% glycerol, 1 mM EDTA. After dialysis glycerol was added to a final concentration of 25% and samples were aliquoted, snap frozen in liquid nitrogen and stored at −80 °C.

### Fluorescence thermal shift assay (FTSA)

Thermal shift assays were performed using CFX96 Touch Real-Time PCR Detection System (Bio-Rad) and CFX Maestro software. The final reaction volume was 25 μL containing 30 μM cyclin B1$^{wt}$ or cyclin B1$^{mut}$ protein mixed with 100-fold diluted SYPRO Orange Dye (100X) in the buffer of 20 mM HEPES (pH 8.0), 600 mM NaCl and 5% glycerol. Melting curves were determined in 96 well plates (Bio-Rad) by continuously increasing the temperature from 25 °C to 95 °C with an increment of 0.1 °C. Melting peaks were determined from the first derivative of the melt curve. For each sample, three duplicates were performed and a control with buffer was included for all the samples.

## UV circular dichroism spectroscopy

Far-ultraviolet CD spectra were recorded on a Chirascan spectro-polarimeter from Applied Photophysics. Experiments were performed in nitrogen atmosphere at room temperature using a 0.1 cm path length quartz cell. Each spectrum was recorded between 260 and 195 nm. The data were collected at a rate of 1 nm/s with a wavelength step of 1 nm and a time constant of 0.5 s. Proteins were dissolved in PBS buffer (137 mM NaCl, 2.7 mM KCl, 8 mM Na2HPO4, and 2 mM KH2PO4, pH 7.4) and a dilution series was performed to improve peak saturation at lower wavelengths.

## Pull-down experiments

25.2 μg purified CDK1$^{MBP}$-cyclin B1$^{wt}$ or CDK1$^{MBP}$-cyclin B1$^{mut}$ mutant was incubated with 28.0 μg separase$^{C2029S}$-securin$^{\triangle160}$ in a 100 μl reaction containing 50 mM HEPES pH 8.0, 100 mM NaCl, 10 mM MgCl$_2$, 5 mM ATP and 5% glycerol. The reaction was performed at RT for 40 min to phosphorylate separase. 80 μl of each reaction was mixed with 40 μl amylose beads and incubated at 4 °C for 1 h. The beads were washed 5 times with washing buffer of 50 mM HEPES pH 8.0, 100 mM NaCl, 10 mM MgCl$_2$, and 5% glycerol, eluted by adding 10 mM maltose and analyzed by SDS−PAGE. For CKS1 pull down, the procedures were similar as described above but without the phosphorylation step.

## Analytic gel filtration

Assembly of separase/CDK1/cyclin B1/CKS1 complex was previously described[29]. Briefly, purified separase$^{C2029S/S1126T}$-securin$^{\triangle160}$ fusion protein was mixed with CDK1/cyclin B1/CKS1 complex at a molar ratio of 1:1.5 in a reaction buffer of 50 mM HEPES pH 8.0, 100 mM NaCl, 10 mM MgCl$_2$, 5 mM ATP and 0.5 mM TCEP. The reaction mixture was incubated at room temperature for 40 min followed by gel filtration on a Superose 6 3.2/300 column (GE Healthcare Life Sciences) in 20 mM HEPES pH 8.0, 150 mM NaCl, 5 mM MgCl$_2$ and 0.5 mM TCEP. For separase$^{C2029S/S1126E}$-securin$^{\triangle160}$ interacting with CDK1/cyclin B1/CKS1, it was performed in a similar way but without the phosphorylation step. Fractions containing separase/CDK1/cyclin B1/CKS1 complexes were analyzed by SDS−PAGE.

## *Xenopus laevis* MII egg extract preparation

For preparation of MII *Xenopus laevis* egg extract, female adult frogs were primed with HCG (chorionic gonadotropin, human, Sigma, 50 U). Primed frogs were injected with a second dose of HCG (500 U) one day before egg-laying and put to separate tanks containing MMR buffer (5 mM HEPES pH7.8 (NaOH), 0.1 mM EDTA, 100 mM NaCl, 2 mM KCl, 1 mM MgCl$_2$, 2 mM CaCl$_2$). The next day, clutches were collected and washed with MMR buffer, thereby apoptotic eggs were removed. Afterwards, eggs were subjected to dejellying for up to 7 min using the dejellying buffer (2% (w/v) L-cysteine, 100 mM KCl, 1 mM MgCl$_2$, pH7.8 NaOH). The dejellying solution was removed by washing three times with CSF-XB buffer (100 mM KCl, 2 mM MgCl$_2$, 0.1 mM CaCl$_2$, 5 mM EGTA, 50 mM sucrose, 10 mM HEPES pH7.7 (KOH)). Dejelled eggs were compacted by centrifugation for 1 min at 200 × $g$ and 1 min at 650 × $g$ and 4 °C in tubes containing 750 μL CSF-XB and 0.1 mg/mL cytochalasin B (sigma). Egg lysis was performed by centrifugation for 10 min at 16,500 × $g$ and 4 °C after removing apoptotic eggs. The cytoplasmic fraction was collected using a syringe and stored at 4 °C together with 10 μg/mL cytochalasin B. The separate egg extracts were tested for MII status using 600 μM CaCl$_2$ and sperm nuclei.

## pT97 antibody validation

For IP experiments, anti-Flag antibody was bound to DynabeadsTM Protein G (Invitrogen) according to manufacturer protocol, 4 μg antibody per 2 μl IVT (In-vitro transcription & translation) was used for binding. Previous to IP, beads were washed 3x with PBST and 1x with PBS. Flag-IP of Flag-Emi2 was performed in CSF-extract arrested in a meiotic state with 1:20 Myc-Emi2$^{CT}$ (aa 491-651, T545A, T551A) and MG262 (100 μM). IVT of Myc-cyc B3 (1:10) and Flag-Emi2 (1:20) constructs were incubated for 30 min at 20 °C. Previous to IP CSF-extract was diluted 1:4 with IP buffer (1x Protease Inhibitor, 1x PhosStop, 20 mM Tris-HCl, pH 7.5, 100 mM NaCl, 10 mM EDTA, 5 mM NaF, 1 mM Na3VO4, 1 mM DTT). IP was performed for 30 min at RT. After IP, beads were washed three times with PBS with 0.025% Tween 20 or two times with PBST and once with PBS. Samples were taken at indicated time points and diluted in 1.5x sample buffer (90 mM Tris HCl pH 6.8, 5% SDS (w/v), 15% Glycerol, 7.5% β-Mercaptoethanol (w/v), Bromphenolblue).

## SDS-PAGE and western blot

Protein samples diluted in 3 x Laemmli buffer (180 mM Tris-Cl pH 6.8, 10% SDS, 30% glycerol, 15% β-mercaptoethanol and bromphenolblue) were heated for 3 min at 95 °C before loading onto SDS-PAGE gels consisting of a stacking gel (375 mM Tris pH 8.8, 0.1% SDS, 8–12% acrylamide/bisacrylamide) and a separation gel (62.5 mM Tris pH 6.8, 0.025% SDS, 5% acrylamide/bisacrylamide). Proteins were separated by size by applying a current of 25 to 60 mA. PageRuler unstained protein ladder (Thermo Fisher) and Pre-stained BenchMark (Invitrogen) were used as markers. For immunoblots, SDS gels were blotted via the wet-blot onto a nitrocellulose membrane (Amersham Protran, 0.45 μm NC, GE Healthcare) at 120 V. Blotted membranes were incubated in milk (1x PBST, 5% milk powder) for 1 h at RT. Next, membranes were incubated in primary antibody overnight, at 4 °C. Afterwards, blots were washed 3x with 1x PBST à 5 min followed by incubation with secondary antibody for 1 h, RT. Blots were again washed 3x with 1x PBST à 10 min, followed by detection of chemiluminescence signals using ECL solution (SuperSignal West Pico Plus, Thermo Fisher) and a LAS-3000 (Fujifilm) imaging system.

## Mass spectrometry analysis of in vitro phosphorylated APC/C complex

Four independent purifications of the APC complex pretreated with lambda phosphatase were subjected to in vitro phosphorylation reactions incubating with CDK1, CSK1 and cyclinB1$^{wt}$ or cyclinB1$^{mut}$. As control the purified lambda phosphatase APC/C was incubated without any kinase. Samples were denatured in 6.8 M Urea and then reduced, alkylated and digested with LysC/Trypsin (protein: enzyme ratio; 50:1). Obtained peptides were concentrated and desalted on C18 StageTips. 100 ng of obtained peptides were separated on a Vanquish™ Neo UPLC System (ThermoFisher Scientific) or EASY-nLC 1200 HPLC system (ThermoFisher Scientific) using a 45 min gradient from 5 to 60% acetonitrile with 0.1% formic acid. All samples were measured as technical triplicates. Data were acquired with a Q-exactive H-FX (ThermoFisher Scientific) in a data-dependent mode acquiring one survey scan (MS scan) and subsequently 15 MS/MS scans of the most abundant precursor ions from the survey scan. The mass range was set to m/z 300 to 1600 and the target value to $3 \times 10^6$ precursor ions with 1.4 Th isolation window. The maximum injection time for purified samples was 28 msec for MS and MS/MS scans. MS scans were recorded with a resolution of 60,000 and MS/MS scans with 15,000. Unassigned precursor ion charge states and singly charged ions were excluded. To avoid repeated sequencing already sequenced ions were dynamically excluded for 30 s. Resulting raw files were processed with the MaxQuant software (version 1.6.14) using the uniport Spodoptera frugipera database adding the proteins used in the in vitro assay (APC subunits, kinases and lambda phosphatase) containing only the proteins present in the samples (APC subunits, kinase and lambda phosphatase)[82]. Oxidation (M) and phosphorylation (STY) were given as variable modifications and carbamidomethylation (C) as fixed modification. A false discovery rate cut-off of 1% was applied at the peptide and protein levels and as well on the phosphorylation peptides. Phospho (STY) and protein groups.txt files were further

processed in Perseus (version 1.6.50)[83]. To obtain a list of high-confident phosphorylation sites of purified APC, phosphopeptides of cyclin B1[wt] were filtered to be present in at least in 2 out of 3 technical replicates and to be found in at least 3 out of 4 independent purifications. Additionally, phosphopeptides were required to be present in >50% of all raw files measured (12 raw files, 4 × 3 technical replicates). Phosphosites present in the control (without kinase) which were not at least two-fold increased after kinase incubation were removed (2 phosphopeptides). After applying the filtering criteria, we obtained 25 phosphopeptides and 28 different phosphosites. The respective cyclinB1[mut] data were filtered in the same manner and then the phosphosites were divided in three groups. (1) cyclinB1[mut] phosphosites which passed the same criteria as the wt (11 phosphosites). (2) Phosphosites which are completely absent in the cyclinB1[mut] samples (12 phosphosites). (3) Phosphosites in the CCC[mut] samples that are at least two-fold reduced and present half or in less of half of the raw files compared to the CCC[wt] (5 phosphosites). Phosphosites were further separated in CDK1 consensus (pS/T P) or other.

## Mass spectrometry analysis of immunoprecipitated APC/C complex

Three independent immunoprecipitations anti-APC3 from HeLa cells expressing either cyclinB1[wt] or cyclinB1[mut] (with knock-down of endogenous cyclinB1) were denatured in 8 M Urea and digested directly on beads as previously described[84]. Obtained peptides were concentrated and desalted on C18 StageTips. Samples were separated on a Vanquish Neo UPLC System (ThermoFisher Scientific™) using a 90 min gradient from 5 to 60% acetonitrile with 0.1% formic acid. Peptides were directly sprayed via a nano-electrospray source in an Orbitrap Exploris™ 480 (ThermoFisher Scientific™). Data were acquired in a data-independent mode acquiring one survey scan (MS scan) with 120 k resolution and subsequently 85 windows with an isolation width of 7.7 The with 1 Dalton overlap from 300 to 1000 m/z at a resolution of 15 k (3 s cycle time, 6 data points on average). The target value was set to 300% for the MS scan and 1000% for the MS/MS scans. The maximum injection time was 45 msec for MS. HCD Collision energy was set to 30%. Resulting raw files were processed with DIA-NN (version 1.8.2) using a Uniprot human database (reference January 2023)[85]. Oxidation (M); M excitation and N-terminal acetylation were given as variable modifications and carbamido-methylation (C) as fixed modification. A false discovery rate cut-off of 1% was applied. The report.pg_matrix.tsv file was further processed in Perseus (version 1.6.50)[83]. Log$_2$ intensity values from the identified APC subunits, UBE2S, CDC20 and cyclinB1[wt] and cyclinB1[mut] were extracted. APC subunit fold changes between the two conditions were calculated by subtracting log$_2$ (cyclinB1[wt] - cyclinB1[mut]) intensity values and further normalized on the ratio of the APC3 bait. Fold changes were plotted in GraphPad Prism.

## Mass spectrometry analysis of immunoprecipitated cyclin B1

In-gel digest was performed as previously described[86]. In short, stained protein bands were cut out and destained. Proteins were reduced via incubation in 100 mM NH$_4$HCO$_3$, 10 mM DTT for 30 min at 56 °C, followed by alkylation via incubation in 100 mM NH$_4$HCO$_3$, 55 mM iodoacetamide for 20 min at 23 °C in the dark. Gel pieces were incubated with 13 ng/μl trypsin (Promega, sequencing grade) in 10 mM 100 mM NH$_4$HCO$_3$, 10% ACN for 2 h on ice, then proteins were digested at 37 °C for 18 h under agitation. Peptides were extracted and samples were dried in a SpeedVac and stored at −20 °C. Dried peptide samples were resuspended in 0.1% TFA for 15 min at 37 °C under agitation and acidified with TFA to a pH <3. Samples were loaded onto equilibrated Pierce C18 SpinTip columns (Thermo, 10 μg) and desalted according to the manufacturers protocol. Eluted peptides were dried in a SpeedVac.

Desalted and dried peptides were resuspended in 5% ACN, 0.1% FA. The peptide amount was estimated measuring the absorbance at 215 nm on a Nanophotometer (Implen) and equal amounts per sample were analysed using an EASY-nLC 1200 UHPLC system (Thermo Fisher Scientific) equipped with a 50 μm × 15 cm C18 Acclaim PepMap RSLC column with 2 μm beads and 100 Å pore size (Thermo Fisher Scientific), connected to a stainless-steel emitter and coupled to a Q-Exactive HF mass spectrometer (Thermo Fisher Scientific) operating in data-independent (DIA) mode. Peptides were separated over a total gradient length of 90 min (after 4 min at 5% ACN the concentration was increased to 35% ACN over 75 min, then to 45% ACN over 5 min followed by a washing step at 80% ACN). Three biological replicates per sample were measured in technical duplicates. MS1 full scans were acquired with a resolution of 120,000 (at 200 m/z) over the range of 300 to 1650 m/z (maximum injection time: 60 ms, AGC target: 3e6). After each full scan, 22 microscans with varying isolation window sizes were conducted with a resolution of 30,000 (maximum injection time: auto, AGC target: 3e6, normalized stepped HCD: 25.5, 27, 30, fixed first mass: 200 m/z). Isolation windows were chosen as described previously[87]. The total DIA cycle time was 2.45 s resulting in 8 data points per peak on average.

Raw data from MS measurements were analysed using Spectronaut (V18, Biognosys). Identification and label-free quantification of proteins was performed in directDIA mode without spectral library. Default settings with normalization but no imputation were used. The Swissprot database UP000005640 (downloaded 07/09/2021 from uniprot.org) was used for the protein search. All statistical analyses of normalized LFQ data were performed using Perseus 1.6.15.0[88]. LFQ values were log2 transformed and only proteins with 6 out of 6 valid values in at least one sample group were kept. Missing values were imputed separately for each measurement from a normal distribution (width = 0.3, downshift = 1.8). Proteins with significantly different abundances between sample groups were identified via ANOVA (S0 = 0.25, permutation based FDR = 0.05) while keeping technical replicate groups preserved for FDR randomization. Significant proteins from the multiple sample test were selected and tested via post-hoc Tukey HSD (FDR 0.05) for differences between sample groups. For hierarchical clustering and heat-map visualization, proteins that were enriched in the wt- control sample over wt+ and mutant samples were filtered out, and then results were filtered for proteins that were enriched in the wt+ over the mutant sample. Transformed protein abundances were z-score normalized and hierarchical clustering was performed using the Euclidean distance with default values in Perseus.

## Statistics and reproducibility

Statistical analyses were performed using Graphpad Prism. If not otherwise stated, all experiments were conducted with at least three independent biological replicates. All unpaired t-tests used here were two-sided.

## Reporting summary

Further information on research design is available in the Nature Portfolio Reporting Summary linked to this article.

## Data availability

All data reported in this paper will be shared by the lead contact upon request. This paper does not report original code. Any additional information required to reanalyze the data reported in this paper is available from the lead contact upon request. The mass spectrometry proteomics data have been deposited to the ProteomeXchange Consortium via the PRIDE[89–91] partner repository with the dataset identifiers PXD048838 and PXD047204. Source data are provided with this paper.

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

## Acknowledgements
This work was supported by funding of the German Research Foundation (DFG, T.B.: 5041140321 and INST 86/1800-1 FUGG; T.U.M.: 532411725 and SFB969/B01, F.S.: 496470458) the LMU Munich's Institutsional Strategy LMUexcellent within the framework of the German Excellence Initiative (T.B.), and the Friedrich-Baur Stiftung (T.B.). A.B. was supported funding of the Swiss National Science Foundation (SNSF, A.B.: TMSGI3_211581), Swiss Cancer Research Foundation (KFS-5453-08-2021 A.B.). We also thank Jörg Mansfeld and Martin Möckel for providing the Flag-APC4 cell line and recombinant proteins, respectively. Support for microscopy was provided by the bioimaging center of the University of Konstanz (BIC).

## Author contributions
A.B. and T.U.M. conceived the study. C.H. performed most experiments. A.H. and J.Y. performed experiments involving recombinant APC/C. R.S. designed and validated the pT97 antibody. P.H. and F.S. performed MS analyses of cyclin B1 IP. N.K. and T.B. performed MS analyses of in vitro phosphorylation assays and APC3 IPs. T.U.M. wrote the manuscript with input of all co-authors.

## Funding

## Competing interests
The authors declare no competing interests.
