## [Transparent Peer Review file · Nature Communications]

Positively charged specificity site in cyclin B1 is essential for mitotic fidelity

Corresponding Author: Professor Thomas Mayer

Version 0:

Reviewer comments:

Reviewer #1

(Remarks to the Author)

I have reviewed the mass spectrometry analysis in the manuscript by Heinzle et al., which overall is proper, however there are several points to be addressed.

1. The mass spectrometry supplementary data should be provided as excel sheets rather than pdf files.
2. In the MS analysis of in vitro phosphorylated APC/C complex, raw files were processed using a reduced database containing only the proteins present in the samples. This is not a proper approach for accurate estimation of false discovery rates in peptide identification. A fasta file with all proteins from the organism plus any additional ones used for the particular experiment should be used for the data processing. Also, this experiment is referred to as phospho-MS, however there is no phosphopeptide enrichment step (e.g. TiO₂). Although there are some filtering criteria applied, there is no statistical analysis of the quantification data (e.g. ttest). It is not clear how peptide quantification was done and how data was normalized. A heatmap of the quantitative data or of discrete values (0=absence, 1=presence) would help the presentation of the data. The mass spectrometer used for this analysis is not shown in the respective methods.
3. Similarly, in MS analysis of immunoprecipitated APC/C complex there is no statistical test applied except the calculation of log₂ (cyclinB1wt - cyclinB1mut).
4. The term "Mass spectrometric" is not very commonly used.

Reviewer #2

(Remarks to the Author)

This manuscript characterizes the potential significance of a phosphate-binding pocket in the Cyclin-dependent kinase 1 (CDK1) activator subunit cyclin B. The investigation follows the recent surprising discovery from a cryo-EM structure of a separate-CDK1 holoenzyme complex that a phosphate on separate contacts positively charged amino acids in cyclin B. Here, the authors develop a knock-down and rescue system in HeLa cells to study the functional significance of the phosphate-binding pocket, and they find that mutation of the pocket leads to defects in the timing of mitotic entry and localization of cyclin B to kinetochores. A mass spectrometry screen leads to biochemical characterization of the anaphase promoting complex (APC/C) as a potential substrate that relies on the phosphate-binding pocket for phosphorylation and its phosphorylation-dependent activity as an E3 ligase.

The results that implicate and detail the significance of the phosphate-binding pocket will be useful to those interested in cell-cycle control and CDKs, biochemical mechanisms regulating mitosis, and potentially cancer biologists considering targeting CDKs. The presence of protein interaction sites on CDK-cyclin-CKS complexes have been well studied and shown to be important mediators of kinase activity, especially in the context of multisite phosphorylation, and localization. For example, the CKS subunit, which also contains a phosphate binding pocket, has already been shown to promote APC/C phosphorylation and activity. The cell model system for studying the cyclin B mutant is well designed and executed and many of the biochemical studies are technically sound. However, for several reasons, publication of the study is rather premature. Some key claims are not sufficiently supported by the completed experiments and data. Moreover, several aspects of the study, including the biochemical characterization of the phosphate pocket activity and the mechanisms underlying the functional consequences of its disruption, remain too underdeveloped for the manuscript to have strong impact. Some specific concerns are outlined below:

1) Several conclusions implicated in the manuscript title are not sufficiently supported in the study. First, it is not clear what is meant by naming the phosphate-binding pocket a “specificity” site and what experimental data support the claim that the pocket directs the kinase to phosphorylate specific sites. There are mass spectrometry results suggesting that more nonconsensus sites can be phosphorylated in the presence of the pocket, but if anything, these observations suggest the kinase is made less specific, so the term is confusing. Second, there are not sufficient data demonstrating that the phosphate-binding pocket is required for mitotic fidelity. Results in Fig. 2 identify a delay in chromosome segregation and that formation of the spindle is less robust upon phosphate-binding pocket mutation, but there are no data showing improper chromosome segregation, aneuploidy, or DNA breaks, which would be needed to implicate mitotic fidelity as it is commonly described in the literature. Perhaps this criticism is a quibble with how “mitotic fidelity” is defined by the authors, but its use in the title may be misleading.

2) The conclusion from data in Fig. 2 that the APC/C is not properly activated is overstated as these experiments do not test APC/C activity. This claim is made once toward the bottom of page 4 (“cyclin B1’s phosphate-binding pocket is critical for the correct function of the mitotic spindle as well as for efficient APC/C activation”) and the top of page 5 (“Since siB1&B2-HeLa mNG-cyclin B1mut cells were inefficient in APC/C activation”). Though shown to be significant, the phenotype in the absence of the SAC (+MPS1 inhibitor) is subtle and there could be other explanations for the delay. Experiments that probe more directly APC/C activity in this system would add significant support toward justifying the conclusion from data in Fig. 2.

3) A weakness of the study is the lack of insight into the mechanism by which loss of phosphate-binding pocket leads to spindle assembly checkpoint activation. Improper localization of cyclin B is clearly demonstrated. But the data in Supp. Fig. 2B for Mad1 binding are not convincing, and there is no further exploration or discussion of whether that interaction is phosphorylation dependent. The significance of the findings would be greatly enhanced if the authors could determine why cyclin B localization to the kinetochore is compromised upon mutation of the phosphate-binding pocket, i.e. which protein-protein interaction is disrupted. Are there any other clues from the mass spectrometry screen?

4) The following conclusion from the data in Fig. 3b is not well justified:
“From these data we concluded that binding of phosphorylated APC/C to CDK1/cyclin B1 is in part mediated by cyclin B1’s phosphate-binding pocket and this interaction is critical for efficient APC/C phosphorylation.” The IP experiment suggests that binding of APC3 is altered such that there is greater preference for a different phosphoform, likely still phosphorylated given the input shown in the overexpression case. Therefore, it is an oversimplification to conclude that the phosphate binding pocket is at all required for association. Moreover, the second part of the conclusion that “the interaction is critical for APC/C phosphorylation” is not supported. Only APC3 is tested here, and the APC3 inputs look similar in the overexpression case. The phosphorylation of APC3 with recombinant enzymes in Fig. 3E (sample 5) also looks similar between the WT and mutant preparations. In sum, while the data that APC/C activity is altered in Figs. 3d and 3f are compelling, there are no convincing data in Fig. 3 that APC/C phosphorylation requires the phosphate-binding pocket or that the overall cyclin B-APC/C complex stability is altered.

5) The mass spectrometry experiment in Figure 4 does suggest that the nature of APC/C phosphorylation is changed, but there are no follow up experiments that validate those results, which diminishes rigor, and there are no experiments that connect any specific phosphorylation differences found in the MS with changes in APC/C activity. More broadly, the study falls short in providing an explanation for why APC/C activity depends on the presence of the phosphate-binding site in cyclin B.

6) Throughout the manuscript, it is implied that the phosphate-binding pocket may stimulate phosphorylation of Cdk1 substrates or change phosphosite specificity, similar to what has been shown for Cks1. However, there are no biochemical assays, beyond the descriptive MS results, which rigorously test this claim.

Reviewer #3

(Remarks to the Author)

This is an interesting and insightful study that reveals more about the anion-binding pocket conserved in B-type cyclins that was shown by the Boland lab to bind to phosphorylated serine 1126 of Separase. The authors present compelling evidence that the anion binding pocket contributes to substrate recognition by B-type cyclin-CDKs, and, like CKS1, can enable recognition of non-consensus CDK sites. They show that this property is important for full activation of the APC/C in mitosis. This study adds anion-pocket binding to CKS1-mediated binding as a mechanism to increase the range of sites recognised by mitotic cyclin-CDKs and thus warrants publication in Nature Communications.

The data are clean and convincing and the experiments have appropriate controls and quantifications. The one caveat is that it is unclear from the Materials and Methods whether the authors tagged Cyclin B1 at its N or C terminus with mNeonGreen. From the description in the text: mNG-Cyclin B1', it appears to be tagged at the N-terminus, which in the experience of this reviewer and other colleagues, greatly slows down its degradation in mitosis. This itself can cause problems in mitosis and is a confounding factor when considering APC/C activation. It will be important to clarify this, and, if necessary, repeat some of the key experiments with a C-terminal-tagged construct.

Reviewer comments:

Reviewer #1

(Remarks to the Author)

The authors have addressed my comments regarding the mass spectrometry analysis described in the manuscript and I have no further concerns.

Reviewer #2

(Remarks to the Author)

In this revised manuscript, the authors added several experiments that improve the study. Specifically, the data in Fig. 2G demonstrating that mutant expression increases the percentage of cells displaying lagging chromosomes support the claim that there is a defect in mitotic fidelity. Also important, the Phos-Tag experiments on APC/C precipitated from cell extracts and experiment with recombinant protein in Fig. 4A strongly support the claim that the phosphate-binding pocket impacts APC/C phosphorylation.

A number of claims have been modified such that they are now fair in light of the supporting data. One exception that should still be addressed is the claim based on data in Fig. 2 that the APC/C is not properly activated. This claim is still made in the results section (lines 232-234) and in the discussion, which points to Fig. 2D as evidence for impaired APC/C activity (lines 461-462). The securin stability data in Fig. S3A is not convincing as the only piece of evidence. It is also not clear why this claim regarding APC/C activity needs to be made based on Fig. 2 data; rather it could be proposed as a potential explanation later in the manuscript after data in Figs. 3 and 4 are reported. The overall message of the study and the need for further evidence would be apparent either way, but the authors would avoid overinterpretation by not making claims about APC/C activity based on Fig. 2 data.

While the lack of mechanistic insights into how the phosphate binding pocket leads to mitotic defects and how changes to APC/C phosphorylation change activity limit the impact of the study, the argument is acceptable that these questions are both challenging to answer and could be considered outside the scope of the current study. Moreover, the finding that the phosphate-binding pocket in Cyclin B has roles in mitosis outside of separase function is an important contribution.

Reviewer #3

(Remarks to the Author)

The authors have satisfactorily addressed my experimental concern but the new data raise a caveat. The decrease in the rate of securin degradation does not point to a major reduction in APC/C activity, and recent data show that cyclin B1 degradation is reduced if its binding to nucleosomes is perturbed (<https://doi.org/10.1038/s44318-024-00194-2>). The authors should discuss the possibility that mutating the phosphate binding pocket may have perturbed nucleosome-dependent APC/C recognition.

First of all, we would like to thank all three reviewers for their positive and constructive comments. Their insightful comments and expert suggestions greatly improved this study. As detailed in our point-by-point response, we have addressed nearly all the issues raised by the reviewers. The new data included in our revised manuscript significantly strengthen our findings. Please find below our point-by-point response addressing their specific comments.

REVIEWER COMMENTS

Reviewer #1 (Remarks to the Author):

I have reviewed the mass spectrometry analysis in the manuscript by Heinzle et al., which overall is proper, however there are several points to be addressed.

We are delighted by the reviewer's enthusiastic feedback and are thankful for her/his comments to improve the quality and readability of the manuscript.

1. The mass spectrometry supplementary data should be provided as excel sheets rather than pdf files.

We agree with the reviewer and provide the supplementary data now as excel sheets.

2. In the MS analysis of *in vitro* phosphorylated APC/C complex, raw files were processed using a reduced database containing only the proteins present in the samples. This is not a proper approach for accurate estimation of false discovery rates in peptide identification. A fasta file with all proteins from the organism plus any additional ones used for the particular experiment should be used for the data processing.

We thank the reviewer for the comment. For the *in vitro* phosphorylation assay, human APC/C, CCC^{wt}, and CCC^{mut} were purified from Sf9 insect cells. We, therefore, rerun the dataset again against the whole proteome of *Spodoptera frugiperda* and human APC/C, CCC^{wt} or CCC^{mut}. We identified 300 phosphopeptides, 181 from insect proteins and 119 of APC/C subunits (before we identified 143 phosphopeptides of APC/C subunits). Using this new dataset, we applied our filtering criteria and, accordingly, created a *new* Fig. 4b and *new* Fig. S4a. Importantly, this new analyses strongly confirms our key finding of the experiment that non-consensus CDK1 phosphosites are lost or reduced in the CCC^{mut} samples.

Also, this experiment is referred to as phospho-MS, however there is no phosphopeptide enrichment step (e.g. TiO2).

We agree with the reviewer that the phrasing is misleading and changed the text from phospho-MS to MS analysis of *in vitro* phosphorylation assays.

Although there are some filtering criteria applied, there is no statistical analysis of the quantification data (e.g ttest). It is not clear how peptide quantification was done and how data was normalized.

We apologize that we did not explain this more in detail. We used DDA acquisition which underlies a certain stochastic sampling. DIA acquisition might have avoided this problem, but we started this project two years ago when DIA just started to be established. For the matter of consistency, we did not want to change the acquisition method. Thus, to be able to have a

reliable identification and quantification of phosphopeptides, we performed the assay 4 times from independent purifications each with 3 technical replicates. In sum, we run 36 rawfiles (12 runs CCC^{wt}, 12 runs CCC^{mut}, 12 runs control without CCC). For quantification of phosphopeptides we used the intensities of the maxquant output table phospho(STY).txt. We wanted to keep only high-confident and robustly quantified phosphopeptides/phosphosites detected under CCC^{wt} conditions for the further analysis. Therefore, we considered only phosphopeptides/phosphosites which were quantified in at least 3 out of 4 replicates, in more than 50% of the raw files and which had a localization probability higher than 95%. After these steps, we remained with 27 phosphosites. Due to the low number and the fact that 10 peptides were completely absent in the CCC^{mut} condition, we did not perform a t-test but sorted the peptides manually in three different categories:

- **Group1:** phosphopeptides are considered as not changed if they pass in the CCC^{mut} data set the same filtering criteria as mentioned above and the intensity of the phosphopeptides changed less than 2 fold in the CCC^{mut} compared to CCC^{wt} assays.
- **Group 2:** phosphopeptides which are in the CCC^{mut} data set at least 2 fold reduced and quantified in half or less than half of the raw files compared to the CCC^{wt}.
- **Group 3:** phosphopeptides which are completely absent in the CCC^{mut} condition.

We made sure that the APC/C subunits and CCC had in all raw files comparable LFQ (label free quantification) intensities by using the quantification of the Maxquant output proteingroups.txt. We, therefore, did not apply any normalization on protein levels (see supplementary table S1). We thank the reviewer for this suggestion because the new analysis and the resulting *new* Fig. 4B and Fig. S4A confirm and further strength our findings.

A heatmap of the quantitative data or of discrete values (0=absence, 1=presence) would help the presentation of the data.

We added a heatmap indicating the three categories to help data visualization (*new* Fig. 4b).

The mass spectrometer used for this analysis is not shown in the respective methods.

The mass spectrometer and methods can be found in the section “material methods”.

3. Similarly, in MS analysis of immunoprecipitated APC/C complex there is no statistical test applied except the calculation of log₂ (cyclinB1^{wt} - cyclinB1^{mut}).

We apologize that we did not explain in detail the logic behind showing the log₂-fold change. For the initial submission, we performed the IP twice with three independent replicates for cyclin B1^{wt} and cyclin B1^{mut}, respectively. A t-test and volcano plot analysis for each experiment revealed that none of the proteins of interest showed a significant change between the wt and mutant IP condition. Yet, despite being not significant, the observed changes were detected in a highly reproducible manner. In response to your comment and to further confirm these results, we repeated the IP a third time in triplicates. This additional experiment provided the same result. We, therefore, feel reassured to report on these changes by showing them as log₂-fold change for the proteins of interest between the wildtype and mutant cyclin B1 IP conditions (*new* Fig. 4C). We hope that the reviewer agrees that this presentation depicts the data as accurately as possible. We changed the text accordingly to emphasize that it is not our

intention to claim significant changes but to report the subtle differences as an observation, which might help in setting up a mechanistic hypothesis.

For the *new* Fig. 4c, we selected the proteins of interest and calculated ratios between the two conditions ($\log_2(\text{cyclin B1}^{\text{wt}} - \text{cyclin B1}^{\text{mut}})$). The bait protein FLAG-APC3 was not equally immunoprecipitated in wt and mutant samples, therefore we then normalized the ratios of the proteins of interest for the ratio of the bait protein and set the \log_2 ratio of FLAG-APC3 to 0.

4. The term “Mass spectrometric” is not very commonly used.

We agree and change the wording to “mass spectrometry analysis”.

Reviewer #2 (Remarks to the Author):

This manuscript characterizes the potential significance of a phosphate-binding pocket in the Cyclin-dependent kinase 1 (CDK1) activator subunit cyclin B. The investigation follows the recent surprising discovery from a cryo-EM structure of a separase-CDK1 holoenzyme complex that a phosphate on separase contacts positively charged amino acids in cyclin B. Here, the authors develop a knock-down and rescue system in HeLa cells to study the functional significance of the phosphate-binding pocket, and they find that mutation of the pocket leads to defects in the timing of mitotic entry and localization of cyclin B to kinetochores. A mass spectrometry screen leads to biochemical characterization of the anaphase promoting complex (APC/C) as a potential substrate that relies on the phosphate-binding pocket for phosphorylation and its phosphorylation-dependent activity as an E3 ligase.

The results that implicate and detail the significance of the phosphate-binding pocket will be useful to those interested in cell-cycle control and CDKs, biochemical mechanisms regulating mitosis, and potentially cancer biologists considering targeting CDKs. The presence of protein interaction sites on CDK-cyclin-CKS complexes have been well studied and shown to be important mediators of kinase activity, especially in the context of multisite phosphorylation, and localization. For example, the CKS subunit, which also contains a phosphate binding pocket, has already been shown to promote APC/C phosphorylation and activity. The cell model system for studying the cyclin B mutant is well designed and executed and many of the biochemical studies are technically sound. However, for several reasons, publication of the study is rather premature. Some key claims are not sufficiently supported by the completed experiments and data. Moreover, several aspects of the study, including the biochemical characterization of the phosphate pocket activity and the mechanisms underlying the functional consequences of its disruption, remain too underdeveloped for the manuscript to have strong impact. Some specific concerns are outlined below:

We thank the reviewer for her/his positive feedback, the thorough analyses of our study and the resulting constructive comments. We have addressed the points raised by the reviewer (see below) and the additional experiments and text changes significantly improved our manuscript.

1) Several conclusions implicated in the manuscript title are not sufficiently supported in the study. First, it is not clear what is meant by naming the phosphate-binding pocket a “specificity”

site and what experimental data support the claim that the pocket directs the kinase to phosphorylate specific sites. There are mass spectrometry results suggesting that more nonconsensus sites can be phosphorylated in the presence of the pocket, but if anything, these observations suggest the kinase is made less specific, so the term is confusing.

We apologize for the confusion regarding the term “specificity site”. As described previously^{1,2}, the ability of CKS proteins to bind pre-phosphorylated proteins can promote the phosphorylation of non-consensus sites by CDK. As alluded by the authors^{1,2}, phosphorylation of non-consensus sites *via* this mechanism can act as “specificity filters” to separate the CDK signal from the one of the many other proline-directed kinases that lack the ability to bind pre-phosphorylated substrates and, hence, are unable to phosphorylate non (S/T)P sites. Based on this concept, we referred to the phosphate-binding pocket of cyclin B1 as “specificity site” because it acts like CKS1 in binding primed substrates, thereby allowing the phosphorylation of non-consensus sites by CDK1/cyclin B1. Yet, to take the justified comment into account we explain in the revised version of our manuscript the rationale behind the term “specificity site” in more detail.

Second, there are not sufficient data demonstrating that the phosphate-binding pocket is required for mitotic fidelity. Results in Fig. 2 identify a delay in chromosome segregation and that formation of the spindle is less robust upon phosphate-binding pocket mutation, but there are no data showing improper chromosome segregation, aneuploidy, or DNA breaks, which would be needed to implicate mitotic fidelity as it is commonly described in the literature. Perhaps this criticism is a quibble with how “mitotic fidelity” is defined by the authors, but its use in the title may be misleading.

We thank the reviewer for this suggestion and performed additional experiments to address this point. As shown in the *new* figure 2G, cells depleted of cyclin B1 and cyclin B2 expressing pocket mutant cyclin B1 show a significantly higher rate of lagging chromosomes compared to wt cyclin B1 expressing cells. No increase in chromosome bridges were observed in mutant cyclin B1 expressing cells (*new* Fig. 2G). Lagging chromosomes are most likely the result of merotelic kinetochore attachments, a defect that can have various causes such as altered microtubule dynamics³⁻⁵. We incorporated these new data into the revised manuscript and appreciate the reviewer's suggestion for these experiments. These novel findings further support our conclusion, as outlined in the manuscript title, that the phosphate-binding pocket of cyclin B1 is crucial for maintaining mitotic fidelity.

2) The conclusion from data in Fig. 2 that the APC/C is not properly activated is overstated as these experiments do not test APC/C activity. This claim is made once toward the bottom of page 4 (“cyclin B1’s phosphate-binding pocket is critical for the correct function of the mitotic spindle as well as for efficient APC/C activation”) and the top of page 5 (“Since siB1&B2-HeLa mNG-cyclin B1mut cells were inefficient in APC/C activation”). Though shown to be significant, the phenotype in the absence of the SAC (+MPS1 inhibitor) is subtle and there could be other explanations for the delay. Experiments that probe more directly APC/C activity in this system would add significant support toward justifying the conclusion from data in Fig. 2.

We appreciate the reviewer for highlighting this point. We agree that the statement regarding APC/C activity was somewhat premature at this stage of the manuscript. In response, we have included new data showing that the degradation of the *bona fide* APC/C substrate, securin, is delayed in reversine-treated cyclin B1/B2-RNAi cells expressing the pocket mutant cyclin B1 compared to those expressing wt cyclin B1 (*new* Fig. S3A). While this effect, similar to the delay in anaphase onset, is subtle, it is highly robust and reproducible. Along with incorporating these new data that support our hypothesis, we have also addressed the reviewer's feedback by revising the text. Specifically, we moderate our conclusion at this stage of the manuscript and acknowledge alternative explanations for the observed delay in anaphase onset. As stated by the reviewer (point 4), our subsequent data regarding diminished APC/C activity are compelling and fully support our initial hypothesis (Figs. 3D and 3F).

3) A weakness of the study is the lack of insight into the mechanism by which loss of phosphate-binding pocket leads to spindle assembly checkpoint activation. Improper localization of cyclin B is clearly demonstrated. But the data in Supp. Fig. 2B for Mad1 binding are not convincing, and there is no further exploration or discussion of whether that interaction is phosphorylation dependent. The significance of the findings would be greatly enhanced if the authors could determine why cyclin B localization to the kinetochore is compromised upon mutation of the phosphate-binding pocket, i.e. which protein-protein interaction is disrupted. Are there any other clues from the mass spectrometry screen?

We agree with the reviewer that gaining more molecular insights into the role of cyclin B1's phosphate-binding pocket in SAC signaling and kinetochore localization would be valuable. Reportedly, CDK1/cyclin B plays a key role in *i*) the proper attachment of kinetochores to microtubules, *ii*) the creation of a checkpoint permissive state, as well as in *iii*) SAC silencing⁶⁻⁹. Thus, altered CDK1/cyclin B function can have pleiotropic effects on SAC signaling making it difficult to tease apart the precise function of CDK1/cyclin B on the different aspects of checkpoint signaling.

The localization of cyclin B1 to unattached kinetochores was reported over a decade ago^{10,11}. Only recently, three studies provided mechanistical insights by describing a direct interaction between cyclin B1 and Mad1¹²⁻¹⁴. However, these studies reached conflicting conclusions about the molecular details of this interaction and its role in checkpoint signaling: Jackman *et al.* showed that Mad1 recruits cyclin B1 to nuclear pores where CDK1/cyclin B1 cooperates with the checkpoint kinase MPS1 to release Mad1 from nuclear pores so that it can be timely recruited to kinetochores to mount an efficient checkpoint response. Alfonso-Pérez *et al.* showed that Mad1 recruits cyclin B1 to unattached kinetochores where it contributes to efficient SAC signaling. In contrast, Allan *et al.* proposed that cyclin B1 is the anchor for Mad1 at the fibrous corona, a transient meshwork that assembles around the outer kinetochore, and, thereby, contributes to SAC signaling. Notably, Mad1 localization to the fibrous corona relied on cyclin B1, but not on the checkpoint proteins Knl1/Bub1/Bub3 or MPS1 activity. As outlined in several recently published review articles trying to reconcile the different findings^{6-8,15}, the confusing results may stem from the complex positive feedback loops that act between the proteins involved affecting their localization and/or activity; a circumstance that makes it challenging to distinguish between causes and consequences. In the case of cyclin B1^{mut}, it gets even more complicated because it causes multiple defects in spindle morphology and chromosome attachment (Figs. 2F and 2G), independently of cyclin B1's role in SAC signaling.

Thus, given the controversies despite the many rigorous experiments conducted, we believe that unraveling the exact molecular details of cyclin B1's pocket in SAC signaling is a separate project beyond the scope of this manuscript. Nevertheless, to address the reviewer's comments we performed a novel experiment shown in the *new* Fig. S2F. Specifically, we treated cyclin B1&B2-RNAi cells expressing wt or pocket mutant cyclin B1 with the MPS1 inhibitor AZ3146. Of note, cells were co-treated with nocodazole to ensure that kinetochores were unattached under all conditions. In cells expressing cyclin B1^{wt}, the Mad1 signal (ratio Mad1/CREST) intensity at kinetochores decreased to roughly 50% within 5 minutes after AZ3146 treatment (*new* Fig. S2F). Thus, these data are in line with Allan *et al.* suggesting that two pools of Mad1 exist one of which localizes to the outer kinetochore in an MPS1-dependent manner and one to the corona in an MPS1-independent manner. Notably, in cells expressing cyclin B1^{mut} the signal intensity of Mad1 at kinetochores was reduced to ~50% compared to wt expressing cells even in the absence of AZ3146. Upon AZ3146 treatment, the Mad1 signal was almost completely lost (*new* Fig. S2F). Based on these and previously published data, we concluded that in cells expressing pocket mutant cyclin B1 Mad1 localizes to the outer kinetochore, but not corona, and this MPS1-dependent localization is lost upon AZ3146 treatment. These new findings providing mechanistic insights into which pool of Mad1 is affected in cells expressing pocket mutant cyclin B1 are discussed in the revised version of the manuscript. As outlined in the above mentioned review articles, a major challenge for future research will be the identification of the cyclin B1 binding partner at the corona.

4) The following conclusion from the data in Fig. 3b is not well justified: "From these data we concluded that binding of phosphorylated APC/C to CDK1/cyclin B1 is in part mediated by cyclin B1's phosphate-binding pocket and this interaction is critical for efficient APC/C phosphorylation." The IP experiment suggests that binding of APC3 is altered such that there is greater preference for a different phosphoform, likely still phosphorylated given the input shown in the overexpression case. Therefore, it is an oversimplification to conclude that the phosphate binding pocket is at all required for association. Moreover, the second part of the conclusion that "the interaction is critical for APC/C phosphorylation" is not supported. Only APC3 is tested here, and the APC3 inputs look similar in the overexpression case. The phosphorylation of APC3 with recombinant enzymes in Fig. 3E (sample 5) also looks similar between the wt and mutant preparations. In sum, while the data that APC/C activity is altered in Figs. 3d and 3f are compelling, there are no convincing data in Fig. 3 that APC/C phosphorylation requires the phosphate-binding pocket or that the overall cyclin B-APC/C complex stability is altered.

We thank the reviewer for pointing this out. As shown previously¹⁶⁻¹⁸, the APC/C is phosphorylated at ~70 serine or threonine residues, and, based on our data, we assume that only a small fraction of these sites are phosphorylated in a pocket-dependent manner. Consequentially, we expect the differences between wt and mutant cyclin B1 to be rather small. Given this context, we performed the following new analyses and experiment to address the reviewer's concern if APC/C phosphorylation indeed requires cyclin B1' phosphate-binding pocket. First, we used a line scan analyses to visualize the SDS-PAGE mobility of APC3 in the input samples from cells expressing either wt or mutant cyclin B1. As shown in the *new* Fig. S3B, the different SDS-PAGE mobility forms of APC3 are shifted to faster migrating forms in

cells expressing pocket mutant cyclin B1 compared to wt expressing cells suggestive of reduced APC3 phosphorylation in the pocket mutant condition. Second, using a stable cell line that expressed Flag-APC4 and mNG-cyclin B1^{wt/mut}, we analyzed the SDS-PAGE mobility of different APC/C subunits using Phos-tagTM gels. For these experiments, we immunopurified the APC/C *via* the Flag tag of APC4. APC4 – irrespective of the Phos-tag conditions applied – did not show a shift in its SDS-PAGE mobility. In contrast, APC1 – like APC3 – showed an enhanced mobility in cyclin B1&B2-RNAi cells expressing mutant cyclin B1 compared to wt expressing cells (*new* Fig. S3B). Third, we performed an *in vitro* APC/C phosphorylation assay using recombinant APC/C and CDK1/CKS1/Cyclin B1^{wt/mut} (CCC^{wt/mut}), both purified from insect cells. Of note, unlike for the *in vitro* kinase assay shown in Fig. 3E, we omitted PLK1 from the reaction to specifically investigate the impact of cyclin B1's pocket on phosphorylation by CDK1. As shown in the *new* Fig. 4A, APC3 *in vitro* phosphorylated by CCC^{mut} shows a clearly faster Phos-tagTM SDS-PAGE mobility than the one phosphorylated by CCC^{wt}. In light of these new data and analyses, we feel confident to state that efficient APC/C phosphorylation depends on the integrity of cyclin B1's phosphate-binding pocket.

As shown in Fig. 3B and the new analyses (Fig. S3B), APC3 co-precipitating with mutant cyclin B1 has a faster SDS-PAGE mobility than the one associated with cyclin B1^{wt}. As aforementioned, pocket integrity is essential for efficient APC/C phosphorylation. Thus, reduced APC/C association with pocket mutant cyclin B1 could be due to the fact that either the phosphorylation sites are missing that mediate CDK1/cyclin B1 binding in a pocket-independent manner, e.g., *via* CKS1, or that the phosphorylation sites are present but cannot bind to pocket mutant cyclin B1. To adequately address the reviewer's comment, we discuss in the revised version of the manuscript these potential scenarios in more detail.

5) The mass spectrometry experiment in Figure 4 does suggest that the nature of APC/C phosphorylation is changed, but there are no follow up experiments that validate those results, which diminishes rigor, and there are no experiments that connect any specific phosphorylation differences found in the MS with changes in APC/C activity. More broadly, the study falls short in providing an explanation for why APC/C activity depends on the presence of the phosphate-binding site in cyclin B.

We agree with the reviewer that we do not provide a molecular explanation of how pocket-dependent phosphorylations of the APC/C affect its activity. Regarding the phosphoregulation of the APC/C, it's important to recall that the initial discovery of APC/C phosphorylation was made more than 20 years ago¹⁹⁻²⁵. Yet, it wasn't until 2016 when the molecular mechanisms underlying APC/C phosphoregulation was provided by seminal publications applying extensive site-directed mutagenesis of individual APC/C subunits, *in vitro* reconstitution experiments and cryo-EM studies^{17,18,26}. Despite this enormous research efforts, our understanding of the molecular mechanism underlying phosphoregulation is still incomplete as highlighted by most recent publications identifying additional layers of complexity in APC/C phospho-regulation²⁷⁻²⁹. Thus, we hope that the reviewer shares our opinion that deciphering the molecular mechanism of pocket-dependent APC/C regulation is a project on its own, going beyond the scope of our manuscript. We view our study – in combination with the co-submitted manuscript of David Morgan's lab – as a pioneering effort that opens up new research avenues by revealing

that the phosphate binding pocket of cyclin B1 is not only crucial for forming a complex with separase but also serves multiple important roles in mitosis.

6) Throughout the manuscript, it is implied that the phosphate-binding pocket may stimulate phosphorylation of Cdk1 substrates or change phosphosite specificity, similar to what has been shown for Cks1. However, there are no biochemical assays, beyond the descriptive MS results, which rigorously test this claim.

We appreciate the reviewer's comment. We have given a lot of thought to how best to address this point and to provide additional data supporting our hypothesis that pocket integrity is critical for sequential substrate phosphorylations involving consensus, but importantly, also non-consensus CDK1 phosphorylation sites. A corollary of our hypothesis is that the phosphorylation pattern of the APC/C, a substrate with >70 phosphorylation sites comprising consensus as well as non-consensus sites with, should be dependent on the integrity of cyclin B1's pocket, while the pattern of a substrate containing a single site should be unaffected. To test this, we performed *in vitro* phosphorylation assays using recombinant CDK1/CKS1/Cyclin B1^{wt/mut} (CCC^{wt/mut}) and apo-APC/C, both purified from insect cells. As a single site substrate, we used a fragment of Emi2 containing a single CDK1 consensus site (Emi2^{NT}; **T**⁹⁷PRVGKK). Notably, both substrates, Emi2^{NT} and apo-APC/C, were present simultaneously in the reaction mixtures. For these assays – unlike the ones shown in figure 3E – we omitted PLK1 to focus exclusively on CCC-mediated phosphorylations. To detect Emi2^{NT} T97 phosphorylation, which does not show a significant shift in its SDS-PAGE mobility upon phosphorylation, we raised and validated an α -phospho-T97 antibody (*new* Fig. S3E). As shown in the *new* Fig. 4A, the phosphorylation pattern of apo-APC/C – but not the phosphorylation efficiency of Emi2^{NT} – is dramatically different in the reaction containing CCC^{mut}, compared to one supplemented with CCC^{wt}. These data strongly support our hypothesis, that the integrity of cyclin B1's phosphate-binding pocket is critical to support sequential substrate phosphorylation. The subsequent MS analyses of the *in vitro* phosphorylation reactions, which was already part of the initial manuscripts, reveals that primarily non-consensus sites are affected.

Reviewer #3 (Remarks to the Author):

This is an interesting and insightful study that reveals more about the anion-binding pocket conserved in B-type cyclins that was shown by the Boland lab to bind to phosphorylated serine 1126 of Separase. The authors present compelling evidence that the anion binding pocket contributes to substrate recognition by B-type cyclin-CDKs, and, like CKS1, can enable recognition of non-consensus CDK sites. They show that this property is important for full activation of the APC/C in mitosis. This study adds anion-pocket binding to CKS1-mediated binding as a mechanism to increase the range of sites recognised by mitotic cyclin-CDKs and thus warrants publication in Nature Communications.

The data are clean and convincing, and the experiments have appropriate controls and quantifications. The one caveat is that it is unclear from the Materials and Methods whether the authors tagged Cyclin B1 at its N or C terminus with mNeonGreen. From the description in the text: mNG-Cyclin B1', it appears to be tagged at the N-terminus, which in the experience of this reviewer and other colleagues, greatly slows down its degradation in mitosis. This itself

can cause problems in mitosis and is a confounding factor when considering APC/C activation. It will be important to clarify this, and, if necessary, repeat some of the key experiments with a C-terminal-tagged construct.

We thank the reviewer for her/his positive feedback and constructive comment. We have addressed the comment as detailed below and feel that the experimental additions have significantly improved the manuscript.

As suggested by the reviewer, we created a novel stable cell line expressing cyclin B1 C-terminally tagged with eGFP and repeated key experiments. As shown in the *new* figures S2C and S2D, these novel data fully confirm our findings using N-terminally tagged cyclin B1. Specifically, cyclin B1&B2-RNAi cells expressing pocket mutant cyclin B1-eGFP show a strong SAC-dependent delay in anaphase onset compared to wt expressing cells (Fig. S2C, left panel), are significantly delayed in anaphase onset in the absence of a functional SAC (Fig. S2C, right panel), and pocket mutant cyclin B1-eGFP fails to localize to kinetochores (Fig. S2D). These novel data strongly support our findings, and we are, therefore, grateful for this suggestion.

REFERENCES:

- 1 Ord, M. *et al.* Multisite phosphorylation code of CDK. *Nat Struct Mol Biol* **26**, 649-658 (2019). <https://doi.org/10.1038/s41594-019-0256-4>
- 2 Valk, E., Örd, M., Faustova, I. & Loog, M. CDK signaling via nonconventional CDK phosphorylation sites. *Mol Biol Cell* **34**, pe5 (2023). <https://doi.org/10.1091/mbc.E22-06-0196>
- 3 de Wolf, B. & Kops, G. Kinetochores Malfunction in Human Pathologies. *Adv Exp Med Biol* **1002**, 69-91 (2017). https://doi.org/10.1007/978-3-319-57127-0_4
- 4 Godek, K. M., Kabeche, L. & Compton, D. A. Regulation of kinetochores-microtubule attachments through homeostatic control during mitosis. *Nat Rev Mol Cell Biol* **16**, 57-64 (2015). <https://doi.org/10.1038/nrm3916>
- 5 Kabeche, L. & Compton, D. A. Cyclin A regulates kinetochores microtubules to promote faithful chromosome segregation. *Nature* **502**, 110-113 (2013). <https://doi.org/10.1038/nature12507>
- 6 Kops, G. & Gassmann, R. Crowning the Kinetochores: The Fibrous Corona in Chromosome Segregation. *Trends Cell Biol* **30**, 653-667 (2020). <https://doi.org/10.1016/j.tcb.2020.04.006>
- 7 Houston, J., Lara-Gonzalez, P. & Desai, A. Rashomon at the kinetochores: Function(s) of the Mad1-cyclin B1 complex. *J Cell Biol* **219** (2020). <https://doi.org/10.1083/jcb.202006006>
- 8 Conde, C. & Gassmann, R. Spindle checkpoint: trapped by the corona, cyclin B1 goes MAD. *Embo j* **39**, e105279 (2020). <https://doi.org/10.15252/embo.2020105279>
- 9 Hayward, D., Alfonso-Pérez, T. & Gruneberg, U. Orchestration of the spindle assembly checkpoint by CDK1-cyclin B1. *FEBS Lett* **593**, 2889-2907 (2019). <https://doi.org/10.1002/1873-3468.13591>
- 10 Bentley, A. M., Normand, G., Hoyt, J. & King, R. W. Distinct sequence elements of cyclin B1 promote localization to chromatin, centrosomes, and kinetochores during mitosis. *Mol Biol Cell* **18**, 4847-4858 (2007). <https://doi.org/10.1091/mbc.e06-06-0539>
- 11 Chen, Q., Zhang, X., Jiang, Q., Clarke, P. R. & Zhang, C. Cyclin B1 is localized to unattached kinetochores and contributes to efficient microtubule attachment and proper chromosome alignment during mitosis. *Cell Res* **18**, 268-280 (2008). <https://doi.org/10.1038/cr.2008.11>
- 12 Alfonso-Pérez, T., Hayward, D., Holder, J., Gruneberg, U. & Barr, F. A. MAD1-dependent recruitment of CDK1-CCNB1 to kinetochores promotes spindle checkpoint signaling. *J Cell Biol* **218**, 1108-1117 (2019). <https://doi.org/10.1083/jcb.201808015>

- 13 Allan, L. A. *et al.* Cyclin B1 scaffolds MAD1 at the kinetochore corona to activate the mitotic checkpoint. *Embo j* **39**, e103180 (2020). <https://doi.org/10.15252/embj.2019103180>
- 14 Jackman, M. *et al.* Cyclin B1-Cdk1 facilitates MAD1 release from the nuclear pore to ensure a robust spindle checkpoint. *J Cell Biol* **219** (2020). <https://doi.org/10.1083/jcb.201907082>
- 15 Cunha-Silva, S. & Conde, C. From the Nuclear Pore to the Fibrous Corona: A MAD Journey to Preserve Genome Stability. *Bioessays* **42**, e2000132 (2020). <https://doi.org/10.1002/bies.202000132>
- 16 Zhang, S. *et al.* Molecular mechanism of APC/C activation by mitotic phosphorylation. *Nature* **533**, 260-264 (2016). <https://doi.org/10.1038/nature17973>
- 17 Qiao, R. *et al.* Mechanism of APC/CCDC20 activation by mitotic phosphorylation. *Proc Natl Acad Sci U S A* **113**, E2570-2578 (2016). <https://doi.org/10.1073/pnas.1604929113>
- 18 Fujimitsu, K., Grimaldi, M. & Yamano, H. Cyclin-dependent kinase 1-dependent activation of APC/C ubiquitin ligase. *Science* **352**, 1121-1124 (2016). <https://doi.org/10.1126/science.aad3925>
- 19 Hall, M. C., Warren, E. N. & Borchers, C. H. Multi-kinase phosphorylation of the APC/C activator Cdh1 revealed by mass spectrometry. *Cell Cycle* **3**, 1278-1284 (2004). <https://doi.org/10.4161/cc.3.10.1153>
- 20 Kraft, C. *et al.* Mitotic regulation of the human anaphase-promoting complex by phosphorylation. *Embo j* **22**, 6598-6609 (2003). <https://doi.org/10.1093/emboj/cdg627>
- 21 Kramer, E. R., Scheuringer, N., Podtelejnikov, A. V., Mann, M. & Peters, J. M. Mitotic regulation of the APC activator proteins CDC20 and CDH1. *Mol Biol Cell* **11**, 1555-1569 (2000). <https://doi.org/10.1091/mbc.11.5.1555>
- 22 Patra, D. & Dunphy, W. G. Xe-p9, a Xenopus Suc1/Cks protein, is essential for the Cdc2-dependent phosphorylation of the anaphase-promoting complex at mitosis. *Genes Dev* **12**, 2549-2559 (1998). <https://doi.org/10.1101/gad.12.16.2549>
- 23 Rudner, A. D. & Murray, A. W. Phosphorylation by Cdc28 activates the Cdc20-dependent activity of the anaphase-promoting complex. *J Cell Biol* **149**, 1377-1390 (2000). <https://doi.org/10.1083/jcb.149.7.1377>
- 24 Shteinberg, M., Protopopov, Y., Listovsky, T., Brandeis, M. & Hershko, A. Phosphorylation of the cyclosome is required for its stimulation by Fizzy/cdc20. *Biochem Biophys Res Commun* **260**, 193-198 (1999). <https://doi.org/10.1006/bbrc.1999.0884>
- 25 Yudkovsky, Y., Shteinberg, M., Listovsky, T., Brandeis, M. & Hershko, A. Phosphorylation of Cdc20/fizzy negatively regulates the mammalian cyclosome/APC in the mitotic checkpoint. *Biochem Biophys Res Commun* **271**, 299-304 (2000). <https://doi.org/10.1006/bbrc.2000.2622>
- 26 Craney, A. *et al.* Control of APC/C-dependent ubiquitin chain elongation by reversible phosphorylation. *Proc Natl Acad Sci U S A* **113**, 1540-1545 (2016). <https://doi.org/10.1073/pnas.1522423113>
- 27 Chia, K. H. *et al.* CDK1-PP2A-B55 interplay ensures cell cycle oscillation via Apc1-loop(300). *Cell Rep* **43**, 114155 (2024). <https://doi.org/10.1016/j.celrep.2024.114155>
- 28 Darling, S. *et al.* The C-terminal disordered loop domain of Apc8 unlocks APC/C mitotic activation. *Cell Rep* **43**, 114262 (2024). <https://doi.org/10.1016/j.celrep.2024.114262>
- 29 Vazquez-Fernandez, E. *et al.* A comparative study of the cryo-EM structures of *Saccharomyces cerevisiae* and human anaphase-promoting complex/cyclosome (APC/C). *Elife* **13** (2024). <https://doi.org/10.7554/eLife.100821>

We would like to begin by thanking all three reviewers for their highly positive feedback. As detailed in our point-by-point response, we have thoroughly addressed the remaining text revisions. Below, you will find our point-by-point responses to their specific comments.

REVIEWER COMMENTS

Reviewer #1 (Remarks to the Author):

The authors have addressed my comments regarding the mass spectrometry analysis described in the manuscript and I have no further concerns.

We are pleased to have successfully addressed the reviewer's comments and to receive her/his recommendation for publication.

Reviewer #2 (Remarks to the Author):

In this revised manuscript, the authors added several experiments that improve the study. Specifically, the data in Fig. 2G demonstrating that mutant expression increases the percentage of cells displaying lagging chromosomes support the claim that there is a defect in mitotic fidelity. Also important, the Phos-Tag experiments on APC/C precipitated from cell extracts and experiment with recombinant protein in Fig. 4A strongly support the claim that the phosphate-binding pocket impacts APC/C phosphorylation.

We appreciate the reviewer's feedback and are glad she/he agrees that the data presented in the revised version provide strong support for our claim that the phosphate-binding pocket is crucial for mitotic fidelity and APC/C phosphorylation.

A number of claims have been modified such that they are now fair in light of the supporting data. One exception that should still be addressed is the claim based on data in Fig. 2 that the APC/C is not properly activated. This claim is still made in the results section (lines 232-234) and in the discussion, which points to Fig. 2D as evidence for impaired APC/C activity (lines 461-462). The securin stability data in Fig. S3A is not convincing as the only piece of evidence. It is also not clear why this claim regarding APC/C activity needs to be made based on Fig. 2 data; rather it could be proposed as a potential explanation later in the manuscript after data in Figs. 3 and 4 are reported. The overall message of the study and the need for further evidence would be apparent either way, but the authors would avoid overinterpretation by not making claims about APC/C activity based on Fig. 2 data.

We agree with the reviewer that it is unnecessary to make the claim of inefficient APC/C activation already at line 232 and to refer at line 461 to figure 2. We changed the text accordingly.

While the lack of mechanistic insights into how the phosphate binding pocket leads to mitotic defects and how changes to APC/C phosphorylation change activity limit the impact of the study, the argument is acceptable that these questions are both challenging to answer and could be considered outside the scope of the current study. Moreover, the finding that the phosphate-binding pocket in Cyclin B has roles in mitosis outside of separase function is an important contribution.

We highly appreciate this positive feedback.

Reviewer #3 (Remarks to the Author):

The authors have satisfactorily addressed my experimental concern but the new data raise a caveat. The decrease in the rate of securin degradation does not point to a major reduction in APC/C activity, and recent data show that cyclin B1 degradation is reduced if its binding to nucleosomes is perturbed (<https://doi.org/10.1038/s44318-024-00194-2>). The authors should discuss the possibility that mutating the phosphate binding pocket may have perturbed nucleosome-dependent APC/C recognition.

We are pleased to have successfully addressed the reviewer's concerns. We agree that the described possibility should be discussed in the revised version of our manuscript and changed the text accordingly.